# Human occupation of northern India spans the Toba super-eruption ~74,000 years ago

Chris Clarkson [1,2,3✉], Clair Harris[1], Bo Li[2,4], Christina M. Neudorf [5], Richard G. Roberts [2,4], Christine Lane[6], Kasih Norman[2], Jagannath Pal[7], Sacha Jones [8,9], Ceri Shipton [10], Jinu Koshy[11], M.C. Gupta[7], D.P. Mishra[7], A.K. Dubey[12], Nicole Boivin[1,3,13,14] & Michael Petraglia [1,3,13✉]

India is located at a critical geographic crossroads for understanding the dispersal of *Homo sapiens* out of Africa and into Asia and Oceania. Here we report evidence for long-term human occupation, spanning the last ~80 thousand years, at the site of Dhaba in the Middle Son River Valley of Central India. An unchanging stone tool industry is found at Dhaba spanning the Toba eruption of ~74 ka (i.e., the Youngest Toba Tuff, YTT) bracketed between ages of 79.6 ± 3.2 and 65.2 ± 3.1 ka, with the introduction of microlithic technology ~48 ka. The lithic industry from Dhaba strongly resembles stone tool assemblages from the African Middle Stone Age (MSA) and Arabia, and the earliest artefacts from Australia, suggesting that it is likely the product of *Homo sapiens* as they dispersed eastward out of Africa.

[1] School of Social Science, University of Queensland, St Lucia, QLD 4072, Australia. [2] Australian Research Council (ARC) Centre of Excellence for Australian Biodiversity and Heritage, University of Wollongong, Wollongong, NSW 2522, Australia. [3] Department of Archaeology, Max Planck Institute for the Science of Human History, Kahlaische Strasse 10, Jena 07745, Germany. [4] Centre for Archaeological Science, School of Earth, Atmospheric and Life Sciences, University of Wollongong, Wollongong, NSW 2522, Australia. [5] Desert Research Institute, Reno, Nevada 89512, USA. [6] Department of Geography, University of Cambridge, Downing Place Cambridge, Cambridge CB2 3EN, UK. [7] Department of Ancient History, Culture and Archaeology, University of Allahabad, Allahabad 211 002 Uttar Pradesh, India. [8] McDonald Institute for Archaeological Research, University of Cambridge, Downing Street, Cambridge CB2 3ER, UK. [9] Office of Scholarly Communication, Cambridge University Library, West Road, Cambridge CB3 9DR, UK. [10] Australian Research Council (ARC) Centre of Excellence for Australian Biodiversity and Heritage, Australian National University, College of Asia and the Pacific, Canberra, ACT 0200, Australia. [11] Department of Ancient History and Archaeology, University of Madras, Chepauk, Chennai 600 005, India. [12] Department of Ancient Indian History, Culture and Archaeology, Banaras Hindu University, Varanasi, India. [13] Department of Anthropology, National Museum of Natural History, Smithsonian Institution, Washington, DC 20560, USA. [14] Department of Anthropology and Archaeology, University of Calgary, Calgary, Canada. ✉email: c.clarkson@uq.edu.au; petraglia@shh.mpg.de

India is a focus of intense debate concerning the timing of the arrival of *Homo sapiens*, the material culture signature of modern human occupation, the nature of replacement of archaic populations, and the impact of the ~74 ka YTT volcanic eruption on hominin populations. While the Indian fossil hominin record is non-existent for this key time period, analysis of mitochondrial DNA of contemporary populations of India indicates that the region was an important geographic stepping stone in the colonisation of Australasia by *Homo sapiens*[1]. At the heart of this debate is the issue of whether *Homo sapiens* arrived in India prior to the YTT event (dated by $^{40}Ar/^{36}Ar$ to $73.88 \pm 0.32$ ka[1] and $75.0 \pm 0.9$ ka[2])[2–10] with a non-microlithic African MSA technology comprised of Levallois and point technology[10–12], or entered the subcontinent around 50–60 ka with Howiesons Poort microlithic technology[13]. While this debate is pivotal to understanding the archaeological signature of modern humans throughout the region, the reality is that very few sites in India are dated to the crucial time period between 80 and 50 ka, hence reliable evidence with which to test competing hypotheses is scarce. Due to the sparse Pleistocene human skeletal record between Africa and South Asia[14,15], the debate over the South Asian record is largely focussed on stone tools and the DNA of modern populations, as well as rare finds such as engraved ostrich egg shell and worked osseous tools from a handful of sites[13].

Here we report detailed descriptions of a rich collection of lithic artefacts from the Dhaba locality, situated on the banks of the Middle Son River in Madhya Pradesh, northern India and comprised of three nearby localities (Dhaba 1, 2 and 3)[16], together with the associated luminescence age estimates. The Dhaba locality provides a detailed archaeological sequence for the Middle Son Valley in a crucial time range of c.80–40 ka, and is positioned chronologically between the early Middle Palaeolithic/Late Acheulean sites of Patpara, Nakjhar Khurd, Sihawal and Bamburi 1, dated to c.140–>104 ka[17,18], and the blade-based 'Upper Palaeolithic' technologies recovered from Baghor formation deposits, previously dated from c.39 ka, although the latter age is problematic[19,20] (see Supplementary Discussion for more detailed discussion and Supplementary Fig. 8 for site locations). In this study, we report infrared stimulated luminescence (IRSL) ages for potassium-rich feldspar (K-feldspar) grains collected from excavated cultural sequences at Dhaba. We use the IRSL ages to frame chronological changes in lithic technology at this site and to place the evidence within the context of the South Asian Palaeolithic and the dispersal of modern humans more broadly[21].

The Dhaba locality is composed of three archaeological excavations (Dhaba 1, 2 and 3) on the north banks of the Son River and west of its confluence with the Rehi River (Figs. 1 and 2)[16]. Each of the three archaeological excavations consisted of a step trench placed into hill slope sediments (Table 1; Figs. 2 and 3). Dhaba 1 (N 24°29′57.6″, E 82°00′35.0″) was selected as the location of densest Middle Palaeolithic surface artefact concentration, with artefacts visibly eroding from sediments at several points up the slope. Dhaba 2 (N 24°29′55.4″, E 82°00′24.5″) and Dhaba 3 (N 24°29′56.1″, E 82°00′22.5″) were selected for excavation due to the existence of eroding accumulations of Middle Palaeolithic artefacts, and a dense concentration of cryptocrystalline microblade and small flake artefacts higher up the slope at Dhaba 3. Excavations at Dhaba 1 and Dhaba 2 are ~600 and ~900 m west of the Rehi-Son River confluence, respectively. The trenches were excavated into colluvial and alluvial sediments overlying Proterozoic sandstone and shale bedrock of the Vindhyan Supergroup[22,23]. Substantial deposits of chemically identified YTT are exposed ~700 m to the east of Dhaba: at Ghogara, on the northern bank

of the Son River[24,25], and in cliff sections on the east bank of the Rehi River[26–28].

The step trenches expose pedogenically altered alluvial sands, silts and clays (Fig. 3, Supplementary Table 1). The tops of the step trenches at Dhaba 1 and 2 are ~16 m above river level. The trench at Dhaba 1 reveals a coarsening-upward sequence of floodplain clays, silts and sands with angular sandstone and shale pebbles, carbonate nodules and rhizoliths. These floodplain sediments overlie angular limestone, sandstone and shale boulders derived from the underlying bedrock (Fig. 3). The trench at Dhaba 2 exposes floodplain clays, silts and sands containing carbonate nodules and a few angular pebbles that overlie shale bedrock. Dhaba 3 is ~1 km west of the Rehi-Son River confluence and consists of a ~3-m-deep trench ~21 m above river level that is dug into the southeastern facing slope of a hill composed of colluvial silts, sands and gravels overlying decomposing sandstone and shale bedrock. The estimated thickness of the colluvial sediments at the top of the hillock is ~5 m. The trench exposes silty sands and pebble gravel with angular sandstone and shale clasts. The hillock is separated from a neighbouring sandstone and shale bedrock ridge, which rises to the west to over 40 m above river level, by south- and southeast-draining gullies that feed into a channel, which, in turn, drains into the Son River. An ~10-m high Holocene terrace composed of sands and silts abuts the north bank of the Son River[16,28]. This terrace overlies large, angular quartzite boulders that are intermittently exposed for ~100 m along the riverfront. Some of these boulders show the removal of large flakes using hard hammer percussion; possibly for the manufacture of quartzite Acheulean cleavers that have been recovered from some sites in the region.

The Dhaba localities together provide evidence of long-term human occupation spanning the last ~80 thousand years. Occupation spans the Toba eruption and the stone tool industry shows no significant change in technology until the introduction of microlithic technology ~48 ka. The lithic industry from Dhaba strongly resembles Middle Stone Age stone tool assemblages from Africa, Arabia and Australia, here interpreted as the product of *Homo sapiens* as they dispersed eastward out of Africa.

## Results

**IRSL chronology.** Thirteen sediment samples from the Dhaba locality were dated using a multiple-elevated-temperature post-infrared IRSL (MET-pIRIR) method[29], described in Methods below. The oldest ages are for Dhaba 1 (Supplementary Table 2) and are stratigraphically consistent with an upper and lower deposit, which mantle the steep slope. The lower unit has IRSL ages of $78.0 \pm 2.9$ and $79.6 \pm 3.2$ ka (Fig. 3a), while the upper unit has IRSL ages of $70.6 \pm 3.9$ and $65.2 \pm 3.1$ ka (Supplementary Table 2). Dhaba 2 was deposited between $55.0 \pm 2.7$ and $37.1 \pm 2.1$ ka (Fig. 3c), while Dhaba 3 has ages of between $55.1 \pm 2.4$ and $26.9 \pm 3.8$ ka (Fig. 3d). The Dhaba sequence, therefore, began accumulating just prior to the YTT event, with only a small likelihood of occuring later, taking the age uncertainties into consideration (p-values of <0.08 and <0.15 assuming a true age for the eruption of 73.88 or 75.0 ka, respectively). Sediment deposition continued until close to the time of the Last Glacial Maximum, making this a unique locality in South Asia with an industrial sequence that stretches from before the YTT event to the microlithic transition.

Interestingly, six glass shards were found at Dhaba 1 in deposits dated to between $79.6 \pm 3.2$ and $65.2 \pm 3.1$ ka (Fig. 3a, see Supplementary Note 1 and Supplementary Table 4), which is consistent with the known date of the YTT event and the widespread presence of YTT in India and the Middle Son Valley[2–10,30]. However, we cannot rule out contamination by

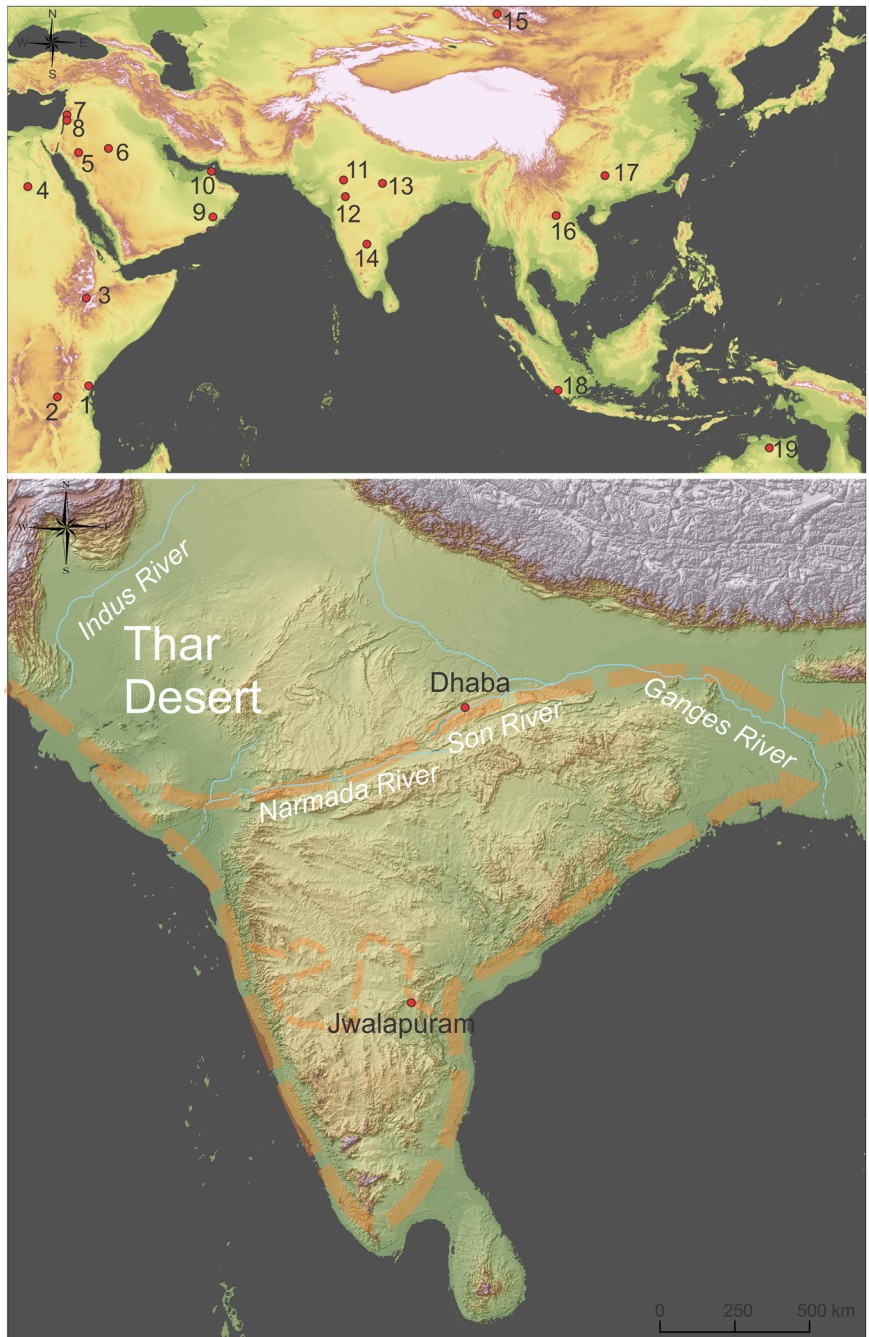

**Fig. 1 Archaeological sites mentioned in the text overlaid on a digitial elevation model of the Eurasian and South Asian landmasses at −60-m sea level consistent with MIS3/4.** Topographic and bathymetric data was obtained from GEBCO 2014 Grid, version 20150318, http://www.gebco.net. Top: Archaeological sites associated with modern humans between Africa and Australia dated >50 ka. 1. Panga ya Saidi; 2. Mumba; 3. Porc Epic; 4. Nazlet Khater; 5. Al Wusta; 6. Jubbah; 7. Qafzeh; 8. Skhul; 9. Dhofar; 10. Jebel Faya; 11. Katoati; 12. Mehtakheri; 13. Dhaba; 14. Jwalapuram; 15. Denisova Cave; 16. Tam Pa Ling; 17. Fuyan Cave; 18. Lida Ajer; 19. Madjedbebe. Bottom: Location of key sites in India and modelled routes of dispersal (dashed orange lines and arrows) from west to east, after Field and colleagues[21].

human agency as a possible source of these few shards at Dhaba 1, given the presence of thick YTT deposits at nearby sites that were visited by some of the same researchers.

**Stone artefacts.** The stone artefact sequence at the three Dhaba excavations spans 55 thousand years, from about 80 to 25 ka, with several distinct pulses in artefact discard (Supplementary Table 1). The sequence is characterised by three major technological phases (Table 1).

The Dhaba 1 assemblage accumulated between around 80 and 65 ka, and contains a predominantly recurrent Levallois core assemblage that includes centripetal, bidirectional and unidirectional recurrent Levallois cores, Levallois flakes, Levallois points, Levallois blades, notches and scrapers (Fig. 4, Supplementary Figs. 5, 7); these tools are made almost exclusively on chert, mudstone and silicified limestone (Fig. 5a). Multiplatform and bidirectional cores and redirecting flakes are also present. Flakes show predominantly strongly radial and weakly radial flake scar patterning, consistent with centripetal core reduction (Fig. 5b,

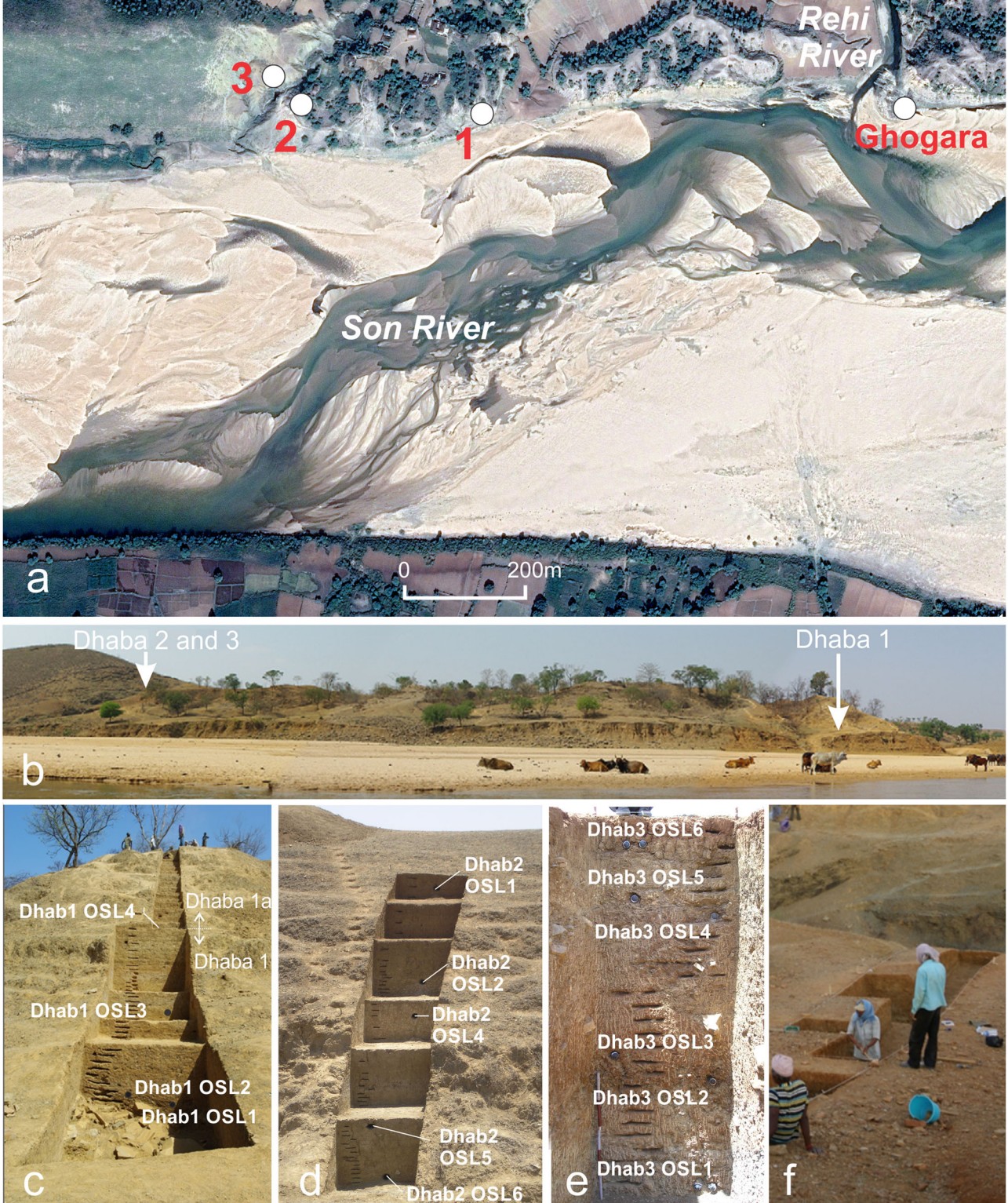

**Fig. 2 Dhaba site location and excavation sections. a** Satellite image of the Son River showing the Rehi confluence and location of the Dhaba excavations (image courtesy of GoogleEarth, image date 11/26/2015, Image © 2019 CNES/Airbus). **b** View of the Dhaba locality from the Middle Son river bed (photo Courtesy of Michael Haslam). **c** Dhaba 1 trench (below), Dhaba 1A trench (above), and locations of IRSL samples. **d** Dhaba 2 trench showing IRSL sample locations. **e** Dhaba 3 section showing IRSL sample locations. **f** View of Dhaba 3 trench during excavation.

Supplementary Figs. 5, 7). Red ochre is also present in the Dhaba 1 assemblage (Fig. 4f, g).

Levallois technology continues to dominate the Dhaba 2 and 3 assemblages between about 55 and 47 ka (strata K and J at Dhaba 3, and strata I to E at Dhaba 2) when artefact deposition peaks. Levallois technology is absent from Dhaba above stratum E, dated to 47.5 ± 2.0 ka. Microlithic technology appears at c.48 ka in stratum D at Dhaba 2 and stratum J in Dhaba 3, with

**Table 1 Composite stratigraphic sequence of counts and proportions of key types through time based on cores, flakes and retouched flakes.**

| Locality | Stratum | IRSL age (ka) | Levallois | % | Microlithic | % | Notches | % | Scrapers | % | Multiplatform | % | Total Artefacts |
|---|---|---|---|---|---|---|---|---|---|---|---|---|---|
| 3 | A | | 0 | 0.00 | 0 | 0.00 | 0 | 0.00 | 1 | 8.33 | 0 | 0.00 | 12 |
| 2 | A | | 0 | 0.00 | 0 | 0.00 | 0 | 0.00 | 0 | 0.00 | 0 | 0.00 | 4 |
| 2 | B | | 0 | 0.00 | 0 | 0.00 | 0 | 0.00 | 0 | 0.00 | 0 | 0.00 | 5 |
| 3 | B | 26.9 ± 3.8 | 0 | 0.00 | 0 | 0.00 | 0 | 0.00 | 0 | 0.00 | 1 | 33.33 | 3 |
| 3 | E | 23.2 ± 2.4 | 0 | 0.00 | 0 | 0.00 | 1 | 1.41 | 1 | 1.41 | 0 | 0.00 | 71 |
| 2 | C | 37.1 ± 2.1 | 0 | 0.00 | 0 | 0.00 | 0 | 0.00 | 0 | 0.00 | 0 | 0.00 | 8 |
| 2 | D | | 0 | 0.00 | 2 | 11.76 | 0 | 0.00 | 0 | 0.00 | 3 | 17.65 | 17 |
| 3 | G | | 0 | 0.00 | 15 | 23.81 | 2 | 3.17 | 1 | 1.59 | 2 | 3.17 | 63 |
| 3 | H | | 0 | 0.00 | 40 | 10.95 | 0 | 0.00 | 3 | 0.82 | 9 | 2.47 | 365 |
| 2 | E | 47.5 ± 2.0 | 22 | 2.74 | 0 | 0.00 | 0 | 0.00 | 2 | 0.25 | 8 | 1.00 | 804 |
| 3 | J | 48.6 ± 2.7 | 5 | 2.02 | 5 | 2.02 | 1 | 0.40 | 0 | 0.00 | 1 | 0.40 | 247 |
| 2 | E/F | | 8 | 2.36 | 0 | 0.00 | 2 | 0.59 | 0 | 0.00 | 0 | 0.00 | 339 |
| 2 | F | 53.9 ± 2.9 | 0 | 0.00 | 0 | 0.00 | 0 | 0.00 | 0 | 0.00 | 0 | 0.00 | 21 |
| 3 | H | 50.8 ± 5.5 | 0 | 0.00 | 0 | 0.00 | 0 | 0.00 | 0 | 0.00 | 0 | 0.00 | 39 |
| 1 | I | 55.0 ± 2.7 | 1 | 0.16 | 0 | 0.00 | 3 | 0.47 | 1 | 0.16 | 3 | 0.47 | 641 |
| 1 | K | 55.1 ± 2.4 | 1 | 0.61 | 0 | 0.00 | 1 | 0.61 | 0 | 0.00 | 1 | 0.61 | 197 |
| 1 | Post-YTT | 65.2 ± 3.1  70.6 ± 3.9 | 4 | 3.03 | 0 | 0.00 | 2 | 1.52 | 1 | 0.76 | 0 | 0.00 | 132 |
| 1 | Pre-YTT | 78.0 ± 2.9  79.6 ± 3.2 | 21 | 2.92 | 0 | 0.00 | 15 | 2.09 | 9 | 1.25 | 6 | 0.83 | 719 |

Microlithic pertains to counts and percentages of blades and microblades and cores as well as backed microliths. Percentages are by stratum in each locality. Totals for Dhaba 3 are from squares 9–13.

microblades, backed artefacts and unidirectional and bidirectional microblade cores all appearing in these strata (Fig. 4h, i, n, o, t u, Supplementary Fig. 6). Quartz is the dominant raw material throughout this microlithic phase, followed by agate (Fig. 5a). Flakes continue to show predominantly centripetal flake scar patterning until the microlithic change (strata 2D and 3G and 3H), when proximal and bidirectional scar patterning becomes the dominant dorsal morphology (Fig. 5b).

By c.37 ka, artefact discard drops dramatically at Dhaba 2 and 3, and very few microlithic artefacts are found after this time (above strata 3C and 2D). Agate and chalcedony are the most common raw materials throughout this final period of site occupation, and flakes show mainly bidirectional and proximal flaking orientations (Fig. 5b).

The broad changes in the proportions of key types through time shown in Table 1 are statistically significant (Pearson chi-square = 2109; $N = 864$; $p = < 0.0005$ one-sided).

## Discussion

The luminescence ages of the Dhaba locality contribute a key missing component in the Middle Son valley chronological sequence, as well as a rare glimpse into the nature of technological change in India between about 80 and 24 ka. The sequence closely mirrors that at Jwalapuram in southern India[31–33], showing a change from recurrent Levallois technology to increasing use of single and multiplatform technology and, then, the manufacture of microlithic assemblages. The technological changes in both the Middle Son and Jurreru River valleys appear to be stepwise and involve broad and statistically significant changes in raw material selection, changing retouch strategies (from scrapers and points to backed artefacts), systematic shifts in core reduction technology, and the introduction of new retouched artefact forms such as backed microliths as Levallois technology disappears[31,32]. Some overlap between Levallois and microlithic technology is also present at Dhaba, with both microlithic and Levallois technology occurring together in stratum J of Dhaba 3 (48.6 ± 2.7 ka) and stratum E of Dhaba 2 (47.5 ± 2.0 ka). The Dhaba sequence presents stratified assemblages spanning the YTT event, and the transition from Levallois to microlithic industries. Other key sites in India also document gradual changes from the Middle Palaeolithic through to the microlithic, such as Bhimbetka[34] and Patne[35], though neither of these sites has been well-dated using modern geochronological techniques and are not known to contain any traces of YTT.

We find that the sequence offered by Dhaba further cements the notion that MSA-like technologies were present in India before and after the YTT eruption[10,31,36]. Lithic technology evolved away from Levallois towards lamellar core reduction systems, and finally saw the introduction of the microlithic (in the form of backed microblades) most likely long after *Homo sapiens* first appeared in the region[31,32].

Recent genetic analyses point to a modern human exit from Africa around 70–52 ka[37,38], in which all contemporary non-African peoples branched off from the same ancestral population that left Africa, possibly with minor genetic contributions from an earlier modern human migration wave[37,39]. Fossil evidence supports earlier dispersals of *Homo sapiens*, with our species present in Greece and the Levant by 200–185 ka[40,41], Arabia by ~85 ka[42], China before ~80 ka[43] and Southeast Asia by 73–63 ka[44], in association with MSA/Middle Palaeolithic technology (where stone artefacts are present). Recent finds from Madjedbebe in northern Australia also document a modern human presence at the eastern end of the 'southern arc' dispersal route by 65 ± 6 ka[45], indicating that groups of *Homo sapiens* likely colonised South Asia prior to this time. The strong connections between

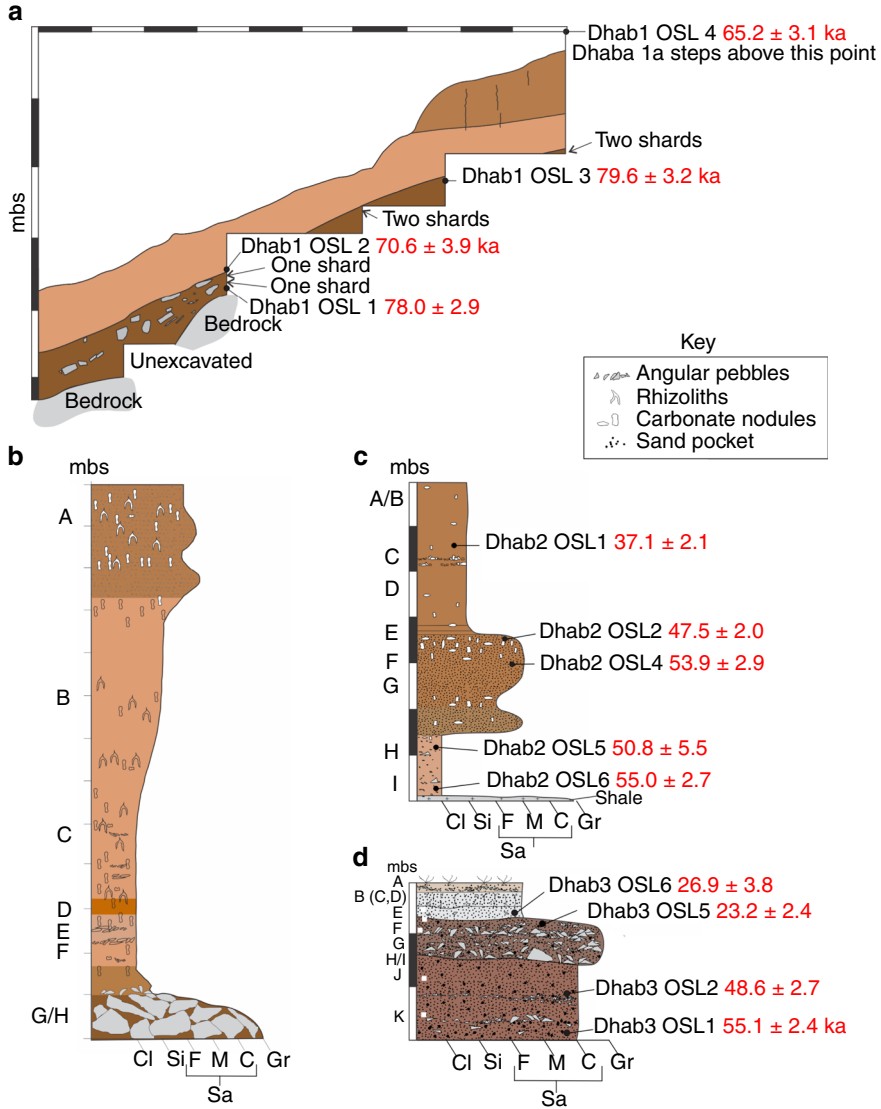

**Fig. 3 Dhaba stratigraphy and chronology. a** Dhaba 1 profile showing location of IRSL ages and cryptotephra shards. The upper 7.88 m of Dhaba 1a are not shown. Note all glass shards are found at or below the boundary between the yellow-brown clay-rich layer and the overlying light yellowish brown silt layer and are bracketed by ages of 78 and 71 ka. **b** Dhaba 1 sediment log. **c** Dhaba 2 sediment log and IRSL ages. **d** Dhaba 3 sediment log and IRSL ages. Cl = clay, Si = silt, Sa = sand (F, M and C are fine, medium and coarse, respectively), Gr = gravel, mbs = metres below surface. Alphabetial references are to stratigraphic layers. Refer to Supplementary Table 1 for detailed sedimentary descriptions.

Aboriginal and South Asian modern human genomes is consistent with dispersal through South Asia[1,46–48] and admixture with Denisovans somewhere along this route[49,50]. The presence of centripetal core and retouched point technology—and the absence of microlithic technology—in northern Australia at c.65 ka makes connections to Southeast Asia, India and East Africa much stronger than previously proposed[11,42]. These technologies co-occur in sites east of Africa dated to between about 100 and 47 ka, suggesting they were likely stepping stones along the southern arc dispersal route[11]. This hypothesis is further supported by quantitative comparisons of core technologies from along this route that point to technological continuity between Africa and Australia[10,11,31]. Modern human dispersal out of Africa, and more importantly east of Arabia, must therefore have taken place before ~65 ka, so cultural and fossil evidence from sites dating to this period will be important for future tests of this hypothesis, notwithstanding the fact that population contractions and turnovers may have also occurred. The Dhaba locality serves as an important bridge linking regions with similar archaeology to the east and west.

## Methods

**Excavation**. Dhabas 1–3 were excavated under permit from the Archaeological Survey of India (No. F.1/36/2008-EE). All trenches was excavated simultaneously by several teams in 1 × 1 m pits arranged as step trenches down the slope at each locality. Excavation trenches were placed in areas where artefacts were eroding from the slope in high density. Dhaba 1 was excavated in 4 lower steps and 1a was excavated in 12 upper steps covering a total elevation of 9–22 m above river level. Dhaba 2 was excavated in six steps covering a total elevation of 21–28 m above river level. The Dhaba 3 trench is located 25–30 m above river level and was 18-m long. Each pit was excavated in spits of ~10-cm depth, with levels taken after each spit using a line level. All excavated sediments were passed through a 5 mm sieve and all artefacts recovered. The weight of the matrix removed during excavation was recorded and all finds were placed in clip seal plastic bags and labelled with provenance details.

**Artefact analysis**. All artefacts were washed and taken to the archaeology laboratory in the Department of Ancient History, Culture and Archaeology at the

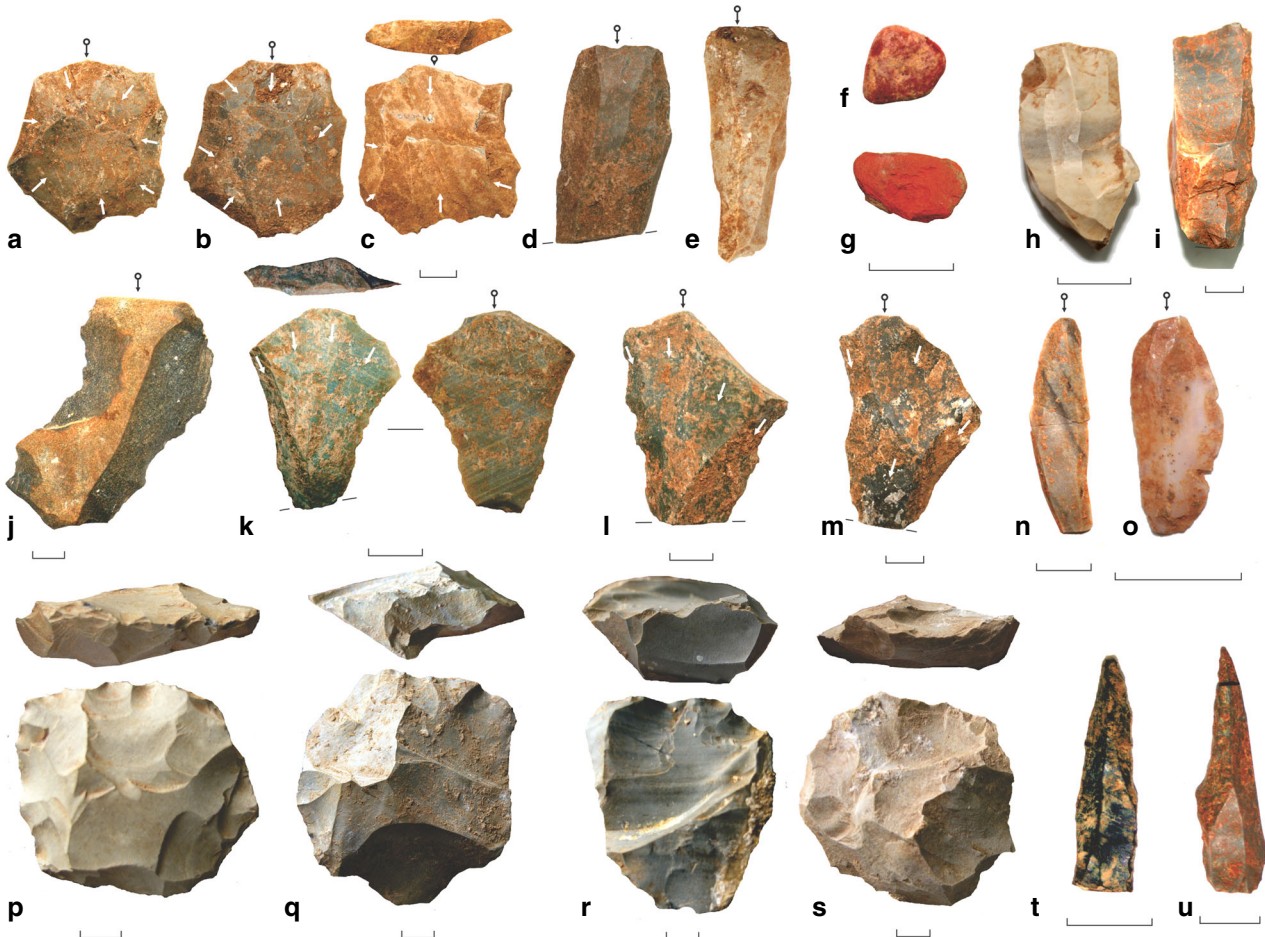

**Fig. 4 Key artefact types at Dhaba from 80 to 25 ka. a–c** Levallois flakes, Dhaba 1 and 2. **d**, **e** Levallois blades, Dhaba 1. **f**, **g** Ochre, Dhaba 1. **h**, **i** Microblade cores, Dhaba 3. **j** Notched scraper, Dhaba 1. **k–m** Levallois points, Dhaba 1 and 2. **n**, **o** Agate and chert microblades, Dhaba 3. **p–s** Recurrent Levallois cores, Dhaba 1–3. **t**, **u** Backed microliths, Dhaba 3. White arrows indicate scar directions. Black arrows with circles indicate impact points.

University of Allahabad for analysis. Each artefact was first classified into technological categories such as core, flake, flaked piece and retouched flake and ascribed typological categories where appropriate. All artefacts were assigned individual specimen numbers, weighed, measured with digital callipers and photographed. All information was entered into a relational database along with detailed provenance information for each artefact. Artefact measurement protocols follow those described in Clarkson et al.[31]. All cores were scanned in three dimensions using a NextEngine laser scanner and a select set of core measurements taken for each[31]. Select artefacts were illustrated using conventional archaeological illustration techniques and protocols.

**IRSL dating**. Sediment samples were collected by hammering opaque plastic tubes (5 cm in diameter) into the cleaned section face. The tubes were removed and wrapped in light-proof plastic for transport to the Luminescence Dating Laboratory at the University of Wollongong. Under dim red laboratory illumination, each sample was treated using standard procedures to extract sand-sized grains of K-feldspar[51,52]. The samples were routinely treated with solutions of HCl acid and $H_2O_2$ to remove carbonates and organic matter, respectively, and then dried. Different grain size fractions in the range of 90–212 μm were obtained by dry sieving, and the K-rich feldspar grains separated using a heavy liquid solution of sodium polytungstate with a density of 2.58 g/cm³. The separated grains were etched using 10% HF acid for ~40 min to clean the surfaces of the grains and reduce the thickness of the alpha-irradiated layer around the grain surface. IRSL measurements of the K-feldspar grains were made on an automated Risø TL-DA-20 reader equipped with IR diodes (875 nm) for stimulation, which delivered ~135 mW/cm² total power to the sample position[53]. Irradiations were carried out within the reader using a ⁹⁰Sr/⁹⁰Y beta source. The IRSL signals were detected using a photomultiplier tube with the stimulated luminescence passing through a filter pack containing Schott BG-39 and Corning 7–59 filters, which provides a blue transmission window (320–480 nm). Aliquots containing several hundred grains (~5 mm in diameter) were prepared by mounting the grains as a monolayer on a 9.8-mm-diameter aluminium disc using "Silkospray" silicone oil as an adhesive.

The dose rates were determined from field measurements of the gamma dose rate, laboratory measurements of the beta dose rate using the sediment samples recovered from each tube hole, and published estimates of the cosmic-ray dose rate and the internal dose rate (due to ⁴⁰K and ⁸⁷Rb contained within the K-feldspar grains). The dosimetry data for all samples are summarised in Supplementary Table 2. The gamma dose rates were measured using an Exploranium GR-320 portable gamma-ray spectrometer, which is equipped with a 3-inch diameter NaI (Tl) crystal calibrated for U, Th and K concentrations using the CSIRO facility at North Ryde. At each sample location, 3–4 measurements of 900 s duration were made of the gamma dose rate at field water content. The external beta dose rate was measured by low-level beta counting using a Risø GM-25-5 multicounter system[54] and referenced to the Nussloch Loess (Nussi) standard[55]. These external components of the total dose rate were adjusted for sample water content, assuming a value of 7 ± 2% for all samples (based on the measured (field) water content of each sample, which ranged from 2 to 5%, and making an allowance for collection of samples during the dry season and partial drying out of the exposed sections prior to sample collection); the assigned uncertainty captures the likely range of time-averaged values for the entire period of sample burial. The minor contribution from cosmic rays was estimated from the burial depth and water content of each sample, and the latitude, longitude and altitude of the Dhaba sites[56]. The internal dose rate was estimated by assuming ⁴⁰K and ⁸⁷Rb concentrations of 13 ± 1% and 400 ± 100 p.p.m., respectively[57–59].

The MET-pIRIR procedure[29,60–62] was applied to determine equivalent dose ($D_e$) of our samples. The IRSL signals of both regenerative and test doses were measured by increasing the stimulation temperature from 50 to 300 °C in steps of 50 °C. A preheat at 320 °C for 60 s was applied after both regenerative and test doses. At the end of the IRSL measurements for each test dose, a 'hot' IR bleach at 325 °C for 100 s was conducted to minimise the residual signal preceding the next measurement cycle. The full experimental procedure is summarised in Supplementary Table 3.

Typical IRSL and MET-pIRIR decay curves and dose response curves (DRCs) for one aliquot of sample Dhab1-OSL4 are shown in Supplementary Fig. 1a, b, respectively. The intensities of the IRSL and MET-pIRIR signals for all the samples

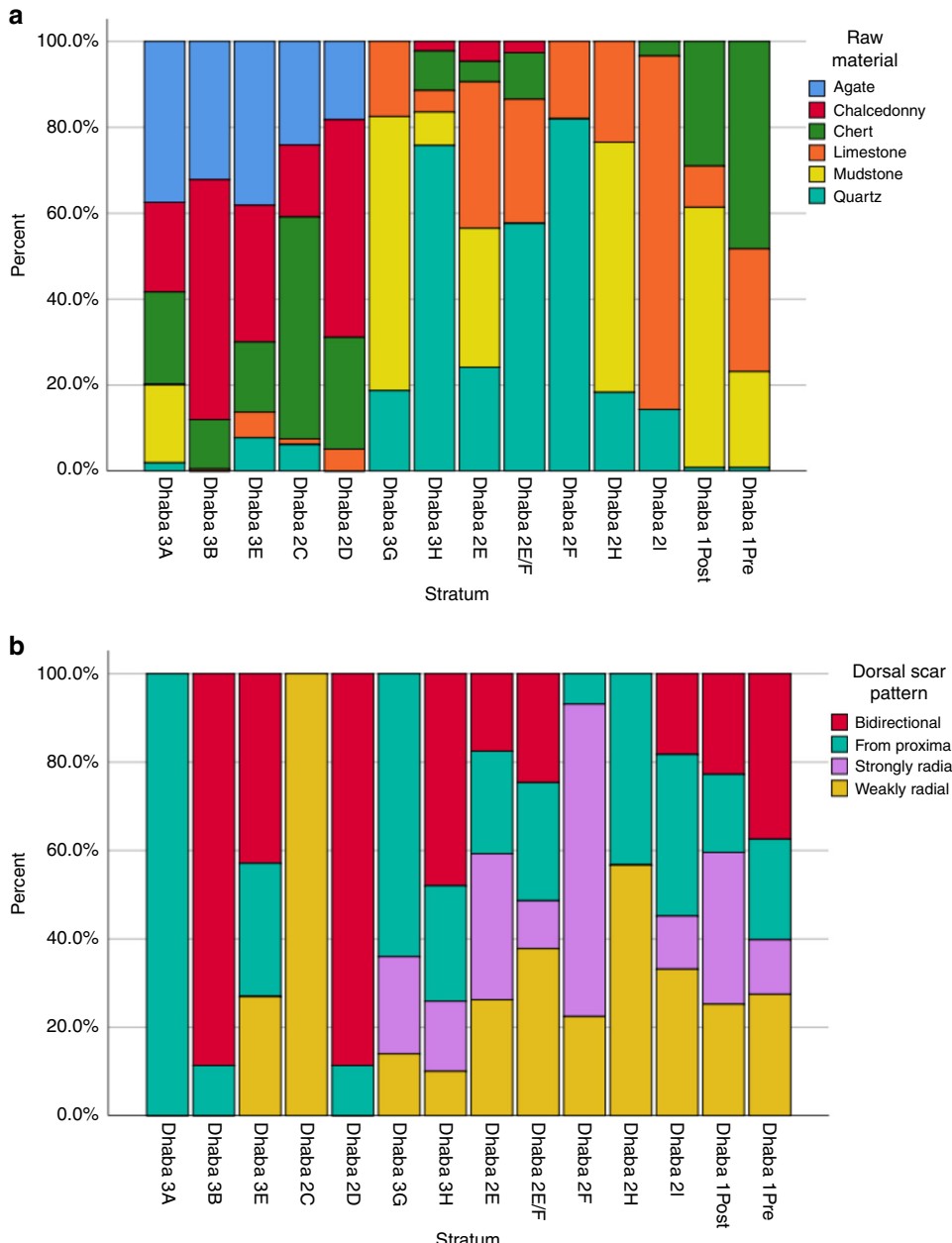

**Fig. 5 Key raw material and technological changes at Dhaba from 80 to 25 ka (left to right = younger to older). a** Raw material changes as a proportion of all artefacts by stratum (Pearson chi-square = 892.4; $N = 9$; $N = 3512$; $p = < 0.005$). **b** Changes in proportions of scar patterning for all complete flakes by stratum (Pearson chi-square = 37.02; df = 9; $N = 797$; $p = < 0.005$ one-sided). Bidirectional scar patterns originate from the proximal (platform) and distal ends of the core or flake. Proximal scar patterns originate from the proximal/platform end only. Strongly radial scar patterns originate in a centripetal pattern around the margins of the flake or core, while weakly radial scar patterns are those with scars that originate from several points around the flake or core circumference but are too few in number (<4) to determine a strong centripetal pattern. Source data are provided as a Source Data file.

are very bright and are on the order of ~$10^5$ counts s$^{-1}$. Different sensitivity-corrected DRCs were observed for the IRSL and various MET-pIRIR signals. These curves were fitted using a single saturating exponential function, which yields characteristic saturation doses of 480, 430, 443, 463, 415 and 308 Gy for the 50, 100, 150, 200, 250 and 300 °C signals, respectively. These results indicate that a natural dose of up to ~800 Gy can be obtained for the Dhaba samples using the MET-pIRIR method.

We tested the applicability of the MET-pIRIR procedure to the Dhaba samples using several routine criteria (e.g., recuperation, recycling ratio, dose recovery, anomalous fading and residual dose)[61,62]. Tests of the recycling ratio and recuperation (i.e., the ratio between the signal responses from a zero regenerative dose and the natural dose) were investigated based on the construction of DRCs for $D_e$ estimation. Recycling ratios for all of the samples fell within the range of 1.0 ± 0.1 and recuperation values were mostly <5%, which are considered acceptable.

For the residual dose test, four aliquots from each of nine samples were bleached by a Dr Hönle solar simulator (model UVACUBE 400) for ~4 h. The residual doses associated with the MET-pIRIR signals were then measured; the results for Dhab2-OSL4 are shown in Supplementary Fig. 1c. The IRSL signal at 50 °C has the smallest residual dose (~2 Gy), which increases as the stimulation temperature is raised. A residual dose of ~18 Gy was obtained for the 250 °C signal, and the highest residual dose (~29 Gy) was observed for the 300 °C signal. The residual doses for the 250 °C signal are summarised for each sample in Supplementary Table 2; the size of the residual dose represents 5–10% of the corresponding $D_e$ value of the 250 °C signal for the Dhaba samples. There is no systematic change in the size of the residual dose with $D_e$ for our samples, which suggests that the non-bleachable traps associated with the residual signal may have been saturated. A simple subtraction of the residual dose from the apparent $D_e$ value may result in underestimation of the true $D_e$ value if the residual signal is

relatively large compared with the bleachable signal[63]. To estimate the proportion of residual signal to bleachable signal for our samples, 12 aliquots of Dhab1-OSL2, Dhab1-OSL3 and Dhab2-OSL1 were heated to 450 °C to empty the source traps associated with the residual and bleachable signals. These aliquots were subsequently given different regenerative doses (165, 330 and 496 Gy) and then bleached using the solar simulator for 4 h before measuring the residual signal using the MET-pIRIR procedure. The measured residual signals from the different regenerative doses were compared with the total regenerative signals at the same doses. The residual signal corresponds to only ~5% of the total signal, which is comparable to the residual dose as a proportion of the measured $D_e$. Given the small size of the residual signal relative to the bleachable signal, the simple dose-subtraction approach should give satisfactory results.

We also tested the validity of the dose-subtraction correction and performance of the MET-pIRIR procedure using a dose recovery test. Four aliquots of sample Dhab2-OSL4 were first bleached by the solar simulator for 4 h and then given a dose of 220 Gy, which was measured as an 'unknown' dose using the MET-pIRIR procedure. The ratios of measured dose to given dose for the IRSL and MET-pIRIR signals are shown in Supplementary Fig. 1d. After correcting for the residual doses shown in Supplementary Fig. 1c, dose recovery ratios of ~0.9 were obtained for the 50 and 100 °C signals, and ratios of $1.02 \pm 0.02$, $1.03 \pm 0.02$, $1.02 \pm 0.02$ and $1.01 \pm 0.03$ for the 150, 200, 250 and 300 °C MET-pIRIR signals, respectively. The results of this dose recovery test suggest, therefore, that the combination of MET-pIRIR and simple dose-subtraction procedures can recover a dose consistent with the known dose given to our samples, so we adopted these procedures to estimate the final $D_e$ values and ages for the Dhaba samples.

Previous studies of pIRIR signals have shown that the anomalous fading rate (g-value) depends on the stimulation temperature, with negligible fading rates observed for MET-pIRIR signals at 200 °C and above[29,60–62]. No fading correction is therefore required for ages estimated from the high-temperature MET-pIRIR signals. To directly test the absence of significant fading for the samples studied here, we conducted anomalous fading tests on K-feldspar grains from samples Dhab2-OSL1 and Dhab3-OSL1 using a single-aliquot measurement procedure similar to that described by Auclair et al.[64], but based on the MET-pIRIR measurement procedure in Supplementary Table 3. The g-values calculated for the IRSL and MET-pIRIR signals (Supplementary Fig. 1e) show that the fading rate is highest for the 50 °C IRSL signal ($3.2 \pm 0.4$ and $4.1 \pm 0.7\%$ per decade for Dhab2-OSL1 and Dhab3-OSL1, respectively) and decreases as the stimulation temperature is raised. The fading rates for the 200 °C signal are <1% per decade and are consistent with zero for the signals measured at 250 and 300 °C, suggesting that negligible fading or non-fading is achieved at the two highest stimulation temperatures.

Based on the above performance tests, the MET-pIRIR procedure was used to measure the $D_e$ values for all samples. The $D_e$ values obtained for each of the MET-pIRIR signals are plotted against stimulation temperature ($D_e$–T plots) for each of the samples from Dhaba 1, 2 and 3 in Supplementary Figs. 2–4, respectively. We also applied a fading correction[65] to the $D_e$ values based on the g-values in Fig. 1e. The fading-corrected $D_e$ values are shown as red squares in Supplementary Figs. 2–4. After applying the fading correction, the fading-corrected $D_e$ values for the 150 and 200 °C signals are consistent with those obtained at higher temperatures (>200 °C), which have negligible fading rates. This further supports our proposition that the MET-pIRIR procedure can access a non-fading signal for the samples studied here and, hence, the $D_e$ values and ages obtained from the elevated temperature signals should be reliable. More importantly, since the signals measured at different temperatures are bleached at significantly different rates (Supplementary Fig. 1c), the consistency in $D_e$ values across a wide range of stimulation temperatures (i.e., 150–300 °C) indicates that our samples had been sufficiently bleached prior to deposition. At lower stimulation temperatures (50 and 100 °C), the $D_e$ values are underestimated, even after correcting for fading, which is consistent with the dose underestimation observed at 50 and 100 °C in the dose recovery test (Supplementary Fig. 1d).

Given the much lower residual dose of the 250 °C signal compared with the 300 °C signal (Supplementary Fig. 1c), we consider the $D_e$ values obtained using the 250 °C signal as the most reliable for the Dhaba samples. The final ages were, therefore, based on the $D_e$ values and associated uncertainties estimated from the 250 °C MET-pIRIR signal (Supplementary Table 2).

**Reporting summary.** Further information on research design is available in the Nature Research Reporting Summary linked to this article.

## Data availability

All relevant data used in this paper are available from the authors. Soil and IRSL dating samples are held in the School of Earth and Environmental Sciences at the University of Wollongong, Australia, and in the Department of Archaeology at the Max Planck Institute for the Science of Human History, Germany. All stone artefacts are held in the Department of Ancient History, Culture and Archaeology at the University of Allahabad, India. The source data underlying Fig. 5 are provided as a Source Data file.

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

## Acknowledgements

We thank the Archaeological Survey of India for permission to conduct the fieldwork, the American Institute of Indian Studies for facilitating this research, and the international team for their contributions to the excavations, especially M. Haslam, A. Crowther and J. Bora. We thank K. Douka for providing comments on a draft of this paper and L. Lewis for conducting cryptotephra lab work. This research was supported by grants from the British Academy (M.P., N.B.), the Leverhulme Trust (M.P., N.B.), the University of Wollongong (B.L.), the European Research Council (M.P.); the Australian Research Council (B.L., C.C., R.G.R.); the McDonald Institute for Archaeological Research (M.P., M. Haslam); and the Max Planck Society (N.B., M.P.).

## Author contributions

J.P., M.P., N.B. and C.C. designed the study. J.P., M.P., C.C., C.H., C.S., J.K., M.C.G., D.P.M., A.K.D. and C.M.N. conducted the fieldwork. C.C., K.N., S.J., B.L., R.G.R. and C.L. wrote the paper. C.C., M.P., N.B. and R.G.R. obtained funding for the study. C.M.N. performed sedimentary analyses. B.L. and R.G.R. carried out IRSL dating. C.L. performed cryptotephra analyses. C.C., C.S. and C.H. analysed the lithic assemblage. C.C., K.N., C.M.N., L.B. and C.H. produced the figures.

## Competing interests

The authors declare no competing interests.
