## [Peer Review File · Nature Communications]

Reviewers' Comments:

Reviewer #1:

Remarks to the Author:

Dear Editor,

Thank you for the opportunity to review Clarkson and colleague's Stone tools and Toba ash demarcate modern human presence in India over 80,000 years. The manuscript presents new data for the occupation of the Indian subcontinent from the Dhaba Locality and related continuity in lithic technology on either side of tephra sherds associated with the Youngest Toba Tuff (YTT) dating elsewhere to ~74 ka. The authors claim that the technological resemblance of the Dhaba stone tool evidence (especially its Levallois components) to African MSA technologies is enough to signify dispersing modern human populations. While I have no doubt about the novelty of this study, I do have reservations about three of the manuscript's central aspects (listed below). I recommend the article be re-reviewed after major revisions.

1. Identifying the volcanic shards

The authors state that the recovered volcanic shard sample size is too small to allow fingerprinting studies. I am not an expert in this field, but is it enough to have the shards date to within a certain time interval to reliably link them to a specific volcanic event (YTT) or do the shards have to be fingerprinted as was the case in the recent study by Smith and colleagues (2018, Humans thrived in South Africa through the Toba eruption about 74,000 years ago)? Their study reports on similar low-frequency tephra shards from which they conducted major and trace element analysis with comparisons to the Volcano Global Risk Identification and Analysis Project database. This enabled the Smith team to make a more secure identification of the YTT shard fingerprint. Clarkson and colleague's manuscript lack these details and it would be worthwhile knowing why they did not perform a similar identification program and what impact the lack of such data might have on their identifications.

Fingerprinting seems even more important at the Dhaba Locality given that the authors state, "the shards are in primary context or were reworked into the overlying deposits, which are several millennia younger." The issues with the Dhaba 1's chronological context are evident in Figure 3 in which tephra shards are marked in association with a date of 70.6 \pm 3.9 straddled in-between two ages of 78 \pm 2.9 (lower) and 79.6 \pm 3.2 (upper). If the YTT identification is correct, then the authors need to explain why their OSL ages do not align with this isochron.

2. Linking the Dhaba archaeology with Homo sapiens dispersals

Recent genetic and fossil evidence has brought further light to the idea that Later Pleistocene hominin diversity was far greater than previously assumed. At least three hominin groups are now known to have occupied parts of Asia at this time, Homo sapiens, Denisovans, and Neanderthals. Whole-genome sequencing of Neanderthals and Denisovans also shows that modern humans contain genetic material from these archaic hominins and that modern humans from the Philippines and New Guinea to China and Tibet inherited roughly 3-5% of their DNA from Denisovans. Considering these findings, can the authors conclusively state that the Dhaba Locality archaeology is the work of Homo sapiens and not another hominin group such as Denisovans?

3. The use of lithic technologies as markers of dispersing human groups

My third comment builds on the comment above and it relates to the validity of using stone tool technologies to track dispersing human groups. Several of the paper's authors have done a commendable job in recent years showing that outdated ideas of a microlith-assisted dispersal of Later Pleistocene humans from Africa is untenable because detailed comparisons of the African and Asian

evidence show little technological resemblance (cf. Lewis, 2017. Early Microlithic Technologies and Behavioural Variability in Southern Africa and South Asia). Yet, in this manuscript, the authors use the presence of Levallois cores and flakes to link the Dhaba Locality assemblage to patterns seen in the African MSA and thereby to dispersing Homo sapiens. The authors should discuss methods by which they could disprove the hypothesis that these technologies represent convergent evolution independent of an African source population. As currently framed, the argument cannot be falsified. In line 150 of the manuscript's supplementary material, the authors state, "A technological analysis of recently excavated artefacts from Patpara reveals the in situ emergence of Levallois reduction". Surely this statement runs directly in the face of the manuscript's central claim that Dhaba Locality's archaeology tracks "Homo sapiens as they dispersed eastward out of Africa."

Below are several more detailed comments for the authors to address:

169: It seems inappropriate to refer to the sequence as "spanning 55 thousand years" and not qualify the statement to the effect that the sequence is highly pulsed and punctuated.

185: If micro is intended to mean small, can the authors explain why microblade cores > 4 cm are called micro (Supp Fig 6) while Levallois cores < 4 cm are not called micro-Levallois (Supp Fig 5)?

189: Can the authors clarify the meaning of proximal scar patterning?

Table 1: Are these core types, flake types, both?

-What comprises the microlithic category in this table? Backed tools, bladelets, small flakes, small cores, bladelet cores? This seems an unnecessary catch-all category.

-Please explain how the percentages are calculated in this table (they don't add to 100 by stratum)?

-Why are backed artefact counts not presented here?

Figure 4: The flake scar patterns (key for identifying Levallois technology) are terribly unclear on tools a, b, and c. The authors should include arrows indicating flake scar directionality on these figures. It is unclear how tool k is a point, it seems to be expanding in plan view, not converging, as one would expect for a pointed tool. The flake scar directionality is also not clear on this tool.

Figure 5 a: Please define strongly and weakly radial so such classifications can be reproduced.

225: Can the authors provide some statistical measures for the "broad changes in raw material selection". For example, is the trend through time significant?

225: I did not see any data on changing retouch strategies, can the authors please highlight where this is?

Reviewer #2:

Remarks to the Author:

Overall, I support this paper. The geoarchaeology looks sound, and the team builds on previous work, and they are all highly experienced in this type of research, particularly in India. Those undertaking the OSL dating are in the top echelons of their field. In particular, the paper strengthens the case for lithic - and possibly population - continuity across the Toba ashfall of 74 ka. In the absence of skeletal evidence, it is the best that can be done at present.

I have a few additional comments:

page 2, line 3: "analysis of modern DNA indicates India was an important geographic stepping stone in the colonisation of Australasia by Homo sapiens". This needs a reference, e.g. Atkinson et al. 2008 Mol. Biol. Evol. 25(2):468–474.

p. 1 and 14, regarding Madjedbebe and Australian colonisation at 65 ka. The inclusion of this is controversial, and does not significantly strengthen their argument for a dispersal pre-80 ka. I can understand why Clarkson wishes to include it as he was the lead author on the Nature paper. Nevertheless, it has been criticised, especially by O'Connell et al. PNAS 2018, and as far as I know, Clarkson et al. have not responded to the suggestion of extensive bioturbation. Personally, I think it should be omitted from the paper.

p. 15: "Modern human dispersal out of Africa must, therefore, have taken place before 65 ka". Two points: first, they should include the recent skeletal evidence from Misliya, Israel, dated at 177-194 ka. Second, the issue here is not "human dispersal out of Africa" because we have the skeletal evidence from Misliya, Skuhl, Qafzeh and Wusta that all show that sapiens was out of Africa before 65 ka. The issue is not dispersal out of Africa, but east of Arabia - it is that part of the Out of Africa debate that is contentious. I suggest rephrase here as suggested.

In the SI, I support their argument that there is considerable spatial variability along rivers such as the Son. Although I have not seen the Son, I have observed similar variability along South Asian rivers and palaeo-rivers, e.g. the Narmada.

Reviewer #3:

Stone tools and Toba ash demarcate modern human presence in India over 80,000 years

Chris Clarkson et al. NCOMMS-19-09215

I was asked to review this paper “*specifically looking for advice on the tephrochronology aspect of this study*” which I have considered. To set this paper in context, the luminescence dating places a series of fluvial deposits from the Son Valley in the time frame of the YTT eruption, and this time is important in hominid migration etc. Many studies have located and Identified YTT in India – the authors cite a limited number of these studies, missing papers in 2013, 2014, and 2017 by Westgate and colleagues which provide analytical data for YTT in India, and mechanisms for recognising YTT.

Terminology: the authors refer to “Toba ash”, “Toba super-eruption” etc. in many places. The Toba Caldera Complex in Sumatra underwent several super-eruptions (OTT, MTT and YTT) thus “Toba ash” is equivocal, and when these terms are used, they should refer to the specific eruption being discussed. There have been suggestion that OTT occurs in India, which have been discounted, but clarity is required when referring to a particular Toba eruption.

Review: Reading the title, which includes “Toba ash”, I was expecting to see samples from a site with a visible tephra layer (which I presumed would be YTT – see comment above) and analyses to confirm the identification of the layer. As I read the paper, the term cryptotephra was used (Line 38, which clearly indicates “*above and below cryptotephra derived from the Toba volcanic super-eruption (Youngest Toba Tuff or YTT) at ~74 ka.*” (again, this is not “**the** Toba super-eruption”, as there were several, but “the YTT super-eruption”). So, not a visible layer, but a cryptotephra layer, but this should provide an isochron, and surely there will be some data or plots of chemistry later in the paper to confirm that headline identification as YTT.

Line 78 finally tells us for the first time about the tephra “layer” describing “*the presence of five tephra glass shards, attributed to the Toba ash*” and again five shards get mentioned on Line 155, but when you look at Figure 3, you count six shards (2+2+1+1) from 4 samples. Is it five or is it six shards? Whatever the number, it is an alarmingly sparse tephra deposit.

Where then is analytical data to confirm the attribution of these few shards to YTT? Are there major element analyses by EPMA which may tie the material to Toba, and perhaps also trace element analyses, the only robust way to identify it as YTT, and distinguish it from OTT (not yet recorded in India, but many argue the possibility that it could be present)? There is no mention of any analyses in the section describing “Cryptotephra shards” (lines 154-166), but mention here of many sites in the vicinity where YTT occurs. The paper continues.....“*The stratigraphic position of the Dhaba 1 shards is consistent with the IRSL ages of 78–80 ka for the lower unit and 65–70 ka for the upper unit, which suggests that the shards are in primary context or were reworked into the overlying deposits*” This sentence implies the shards have been identified and are now in the correct stratigraphic position (based on the OSL ages), but where is the data to show these are YTT?

There is a hint of circularity to this – these shards, which the authors have already said are YTT, are in the right place, because the ages tell us they are Toba? As yet, no analytical data has been presented to suggest these are even rhyolite, let alone material derived from Toba, or even specifically YTT.

In **Methods**, for Cryptotephra (Lines 263 – 271), again no data are represented to justify the assignment to YTT, but the consistency of the ages of the sediments with the eruption age of YTT is used to justify the stratigraphic context, and assignment of a YTT origin. Again, no evidence is presented to establish the identity of the glass by methods which are extensively used in tephra and cryptotephra studies.

Eventually, residing in the **Supplementary information**, you find some more detail of the “*Cryptotephra investigation*”. Standard methods were used to prepare and separate samples hopefully containing glass, and mounted to identify shards microscopically. A few glass shards were found in total, but crucially the glass shards were not analysed “*because they were too low in abundance*”. In the S.I. the paper states “*our correlation to the YTT is based upon the demonstrated presence of YTT within the Son river valley*” – so not an assignment based on compositional data or any other measureable parameter (perhaps they could try RI data which would at least tell you they are/are not rhyolitic), but assumed to be YTT simply because there is some near-by. The OSL ages and the “visual appearance” of the shards are used back this up. The authors also cite the fact that no other tephra apart from YTT has been identified in the area, as further justification (but miss 3 or 4 papers which are post-2011 relating to YTT in India).

The “recognition” of YTT in these samples is thus based on an assumption, and is not identification, and should not be used. Further material should have been prepared so enough glass could be obtained for analysis (this is presumably still possible?), and then (and only then) can a reliable identification be attempted. The headline presence of “Toba tephra” in the title is misleading – and whilst it I would not be at all surprised if this does turn out to be YTT, at present there is no evidence whatsoever to support this.

Summary: Any mention of a Toba/YTT assignment to the cryptotephra should be removed from the paper entirely; it is misleading and not proven. In the context of other sites in the area where there are clear layers of tephra the title leads you to imagine something that is not there, and these misconceptions stick – in 2 years’ time I can imagine a paper appearing which quotes “Clarkson et al. (2019) identified YTT in sediments from Dhaba, India”and there it is, forever entrenched in the literature, when there has actually been no robust identification of the glass.

A more open approach should be adopted in the paper, along the lines of “Five [*or is it six?*] uncharacterised, colourless [*I’m not sure if they state the colour*], platy volcanic glass shards were recovered from four samples taken for luminescence dating. These were at very low concentrations (<1 shard/cm³), and it was not possible to analyse these. As yet, their source volcano cannot be confirmed”, and no further speculation about what they are should be made. It is appropriate to refer to YTT in many other sites across India, but the reader can be left to speculate on whether these are the same – it should not be fed to them without proper evidence to support it. Furthermore, “and Toba ash” should not appear in the title, nor Toba/YTT etc in the abstract or text, figures etc., instead “unidentified glass shards” should be used until it can be proved using standard analytical approaches that these are indeed from the YTT eruption, or not.

Recommendation: The paper needs modification to remove any mention of these shards providing a justification for the stratigraphic context of the artefacts, and needs to be rephrased throughout so as not to indicate their provenance, until this is confirmed by appropriate analytical means.

Reviewer 1

Thank you for the opportunity to review Clarkson and colleague's Stone tools and Toba ash demarcate modern human presence in India over 80,000 years. The manuscript presents new data for the occupation of the Indian subcontinent from the Dhaba Locality and related continuity in lithic technology on either side of tephra sherds associated with the Youngest Toba Tuff (YTT) dating elsewhere to ~74 ka. The authors claim that the technological resemblance of the Dhaba stone tool evidence (especially its Levallois components) to African MSA technologies is enough to signify dispersing modern human populations. While I have no doubt about the novelty of this study, I do have reservations about three of the manuscript's central aspects (listed below). I recommend the article be re-reviewed after major revisions.

1. Identifying the volcanic shards

The authors state that the recovered volcanic shard sample size is too small to allow fingerprinting studies. I am not an expert in this field, but is it enough to have the shards date to within a certain time interval to reliably link them to a specific volcanic event (YTT) or do the shards have to be fingerprinted as was the case in the recent study by Smith and colleagues (2018, Humans thrived in South Africa through the Toba eruption about 74,000 years ago)? Their study reports on similar low-frequency tephra shards from which they conducted major and trace element analysis with comparisons to the Volcano Global Risk Identification and Analysis Project database. This enabled the Smith team to make a more secure identification of the YTT shard fingerprint. Clarkson and colleague's manuscript lack these details and it would be worthwhile knowing why they did not perform a similar identification program and what impact the lack of such data might have on their identifications.

Fingerprinting seems even more important at the Dhaba Locality given that the authors state, "the shards are in primary context or were reworked into the overlying deposits, which are several millennia younger." The issues with the Dhaba 1's chronological context are evident in Figure 3 in which tephra shards are marked in association with a date of 70.6 \pm 3.9 straddled in-between two ages of 78 \pm 2.9 (lower) and 79.6 \pm 3.2 (upper). If the YTT identification is correct, then the authors need to explain why their OSL ages do not align with this isochron.

2. Linking the Dhaba archaeology with Homo sapiens dispersals

Recent genetic and fossil evidence has brought further light to the idea that Later Pleistocene hominin diversity was far greater than previously assumed. At least three hominin groups are now known to have occupied parts of Asia at this time, Homo sapiens, Denisovans, and Neanderthals. Whole-genome sequencing of Neanderthals and Denisovans also shows that modern humans contain genetic material from these archaic hominins and that modern humans from the Philippines and New Guinea to China and Tibet inherited roughly 3-5% of their DNA from Denisovans. Considering these findings, can the authors conclusively state that the Dhaba Locality archaeology is the work of Homo sapiens and not another hominin group such as Denisovans?

3. The use of lithic technologies as markers of dispersing human groups

My third comment builds on the comment above and it relates to the validity of using stone tool technologies to track dispersing human groups. Several of the paper's authors have done a commendable job in recent years showing that outdated ideas of a microlith-assisted dispersal of Later Pleistocene humans from Africa is untenable because detailed comparisons of the African and Asian evidence show little technological resemblance (cf. Lewis, 2017. Early Microlithic Technologies and Behavioural Variability in Southern Africa and South Asia). Yet, in this manuscript, the authors

use the presence of Levallois cores and flakes to link the Dhaba Locality assemblage to patterns seen in the African MSA and thereby to dispersing Homo sapiens. The authors should discuss methods by which they could disprove the hypothesis that these technologies represent convergent evolution independent of an African source population. As currently framed, the argument cannot be falsified.

In line 150 of

the manuscript's supplementary material, the authors state, "A technological analysis of recently excavated artefacts from Patpara reveals the in situ emergence of Levallois reduction". Surely this statement runs directly in the face of the manuscript's central claim that Dhaba Locality's archaeology tracks "Homo sapiens as they dispersed eastward out of Africa."

Below are several more detailed comments for the authors to address:

169: It seems inappropriate to refer to the sequence as "spanning 55 thousand years" and not qualify the statement to the effect that the sequence is highly pulsed and punctuated.

185: If micro is intended to mean small, can the authors explain why microblade cores > 4 cm are called micro (Supp Fig 6) while Levallois cores < 4 cm are not called micro-Levallois (Supp Fig 5)?

189: Can the authors clarify the meaning of proximal scar patterning?

Table 1: Are these core types, flake types, both?

-What comprises the microlithic category in this table? Backed tools, bladelets, small flakes, small cores, bladelet cores? This seems an unnecessary catch-all category.

-Please explain how the percentages are calculated in this table (they don't add to 100 by stratum)?

-Why are backed artefact counts not presented here?

Figure 4: The flake scar patterns (key for identifying Levallois technology) are terribly unclear on tools a, b, and c. The authors should include arrows indicating flake scar directionality on these figures. It is unclear how tool k is a point, it seems to be expanding in plan view, not converging, as one would expect for a pointed tool. The flake scar directionality is also not clear on this tool.

Figure 5 a: Please define strongly and weakly radial so such classifications can be reproduced.

225: Can the authors provide some statistical measures for the "broad changes in raw material selection". For example, is the trend through time significant?

225: I did not see any data on changing retouch strategies, can the authors please highlight where this is?

Please find our responses to each main point below.

Tephra geochemical fingerprinting

The authors state that the recovered volcanic shard sample size is too small to allow fingerprinting studies. I am not an expert in this field, but is it enough to have the shards date to within a certain time interval to reliably link them to a specific volcanic event (YTT) or do the shards have to be fingerprinted as was the case in the recent study by Smith and colleagues (2018, Humans thrived in South Africa through the Toba eruption about 74,000 years ago)? Their study reports on similar low-frequency tephra shards from which they conducted major and trace element analysis with

comparisons to the Volcano Global Risk Identification and Analysis Project database. This enabled the Smith team to make a more secure identification of the YTT shard fingerprint. Clarkson and colleague's manuscript lack these details and it would be worthwhile knowing why they did not perform a similar identification program and what impact the lack of such data might have on their identifications.

We are confident of the correlation of the six tephra glass shards found in the Dhaba 1 sequence to the YTT, even without chemical analyses, because all Middle Son tephra deposits, indeed all Late Pleistocene tephra deposits studied in India, have been correlated to the YTT. This is noted and reviewed in many publications including our own. Indeed, recent elemental fingerprinting studies by Pearce et al. 2014 and 2019 have shown that all known tephra reported from India is the 75 ka YTT. There are no other contemporary Late Pleistocene volcanic sources believed to be able to deposit volcanic ash in this region.

Most significantly, an extensive deposit of YTT is located only 700m to the east of the Dhaba site at Ghogara. Ghogara's YTT deposit has been extensively published and chemically fingerprinted by our team and others. As the cryptotephra shards at Dhaba have also been bracketed by luminescence ages, there is great confidence in assigning them to the YTT.

We recognised that Smith et al. (2018) were able to isolate and analyse ultra-distal YTT shards from their archaeological sedimentary record. The task of doing this in a fluvial deposit dominated by siliciclastic silts and sands, of similar density to the tephra glass shards, is somewhat greater than in a sandy calcareous deposit, where the surrounding material may largely be removed using a weak acid wash. Regardless, it took those authors more than a year to achieve their analyses (note co-author Lane was involved in that study). Whereas their discovery of YTT in Southern Africa was a first, and several hundred kilometres from any other known deposit, at Dhaba 1 we have a clear chronological, sedimentological and tephrostratigraphical association of the glass shards found in Dhaba 1 with the nearby YTT deposits at Ghogara, just 700 m away.

We have amended the cryptotephra section to read:

"Analyses of sediments collected from the units at Dhaba 1 deposited before and after the YTT event revealed the presence of six volcanic glass shards (Supplementary Table S4). Although we were unable to chemically characterise these shards, volcanic ash recovered from the extensive Ghogara tephra deposits ~700 m to the east contained glass shards and biotite minerals that have been chemically fingerprinted as YTT³⁰, and directly dated by ⁴⁰Ar/³⁹Ar to 72.7 ± 4.1 ka². YTT has been geochemically fingerprinted at numerous locations throughout India²⁻⁹, including at the archaeological site of Jwalapuram¹⁰, with all reported occurrences of Toba tephra across India linked unequivocally to the ~74 ka event⁷⁻⁹. The stratigraphic position of the Dhaba 1 shards between the lower unit (80–78 ka) and the upper unit (71–65 ka) is consistent with them also being of YTT age."

Tephra chronology

The issues with the Dhaba 1's chronological context are evident in Figure 3 in which tephra shards are marked in association with a date of 70.6 ± 3.9 straddled in-between two ages of 78 ± 2.9 (lower) and 79.6 ± 3.2 (upper). If the YTT identification is correct, then the authors need to explain why their OSL ages do not align with this isochron.

This is a misinterpretation of the stratigraphic drawing in Figure 3. The tephra shards are in all cases found at the top of the dark brown stratigraphic layer which slopes downhill. In the lower, (downslope) part of the section, the tephra shards are bracketed by ages of 70.6 ± 3.9 and 78 ± 2.9 , and in the upper section by ages of 79.6 ± 3.2 and 65.2 ± 3.1 ka. The tephra shards presumed to be

YTT and dating to around 74ka are therefore in correct stratigraphic position between these ages. Reviewer 1 is confused because there are 2 shards from the mid-slope with no associated ages. These fall in the upper part of the lower brown unit, as do all the other cryptotephra shards. It is incorrect of Reviewer 1 to state that these fall midway between the ages of 70 and 65ka, as those ages are all in the overlying light brown unit and do not bracket the tephra shards. We can confidently conclude that the tephra shards at Dhaba fall between ages of 78 and 65ka and are in the correct stratigraphic position, and thus age, to by YTT. The fact that we have chemically fingerprinted large YTT deposits located only 700m from the site indicates this is highly likely to also be YTT.

We have added the following text to the caption of Figure 4:

“Note all glass shards are found at or below the boundary between the yellow-brown clay-rich layer and the overlying light yellowish grow silt layer and are bracketed by ages of 78 ka and 71 ka.”

Linking the Dhaba archaeology with Homo sapiens dispersals

Recent genetic and fossil evidence has brought further light to the idea that Later Pleistocene hominin diversity was far greater than previously assumed. At least three hominin groups are now known to have occupied parts of Asia at this time, Homo sapiens, Denisovans, and Neanderthals. Whole-genome sequencing of Neanderthals and Denisovans also shows that modern humans contain genetic material from these archaic hominins and that modern humans from the Philippines and New Guinea to China and Tibet inherited roughly 3-5% of their DNA from Denisovans. Considering these findings, can the authors conclusively state that the Dhaba Locality archaeology is the work of Homo sapiens and not another hominin group such as Denisovans?

We cannot conclusively demonstrate that these stone artefacts were made by modern humans. However, our comparisons of lithic technology, assemblage composition and the pattern of technological change between Africa and Australia along the southern arc route all point to a very close similarity between assemblages created by modern humans in Africa, the Levant and Arabia, as well as sites to the east that are not found with fossil evidence in India, SE Asia and Australia but are attributed to modern humans. This is not the case in comparisons with Neanderthal assemblages which are quantitatively different (Clarkson et al. 2012; Petraglia et al. 2007; Clarkson 2014). Our working hypothesis is that the Dhaba assemblages are the product of modern humans. The hypothesis will be tested through collection of more data and fossil finds in the future.

We have added a sentence to the discussion referring to these quantitative studies:

“These technologies co-occur in sites east of Africa dated to between about 100 and 47 ka, suggesting they were likely stepping stones along the southern arc dispersal route¹¹. This hypothesis is further supported by quantitative comparisons of core technologies from along this route that point to technological continuity between Africa and Australia^{10,11,31}.”

The use of lithic technologies as markers of dispersing human groups

My third comment builds on the comment above and it relates to the validity of using stone tool technologies to track dispersing human groups. Several of the paper’s authors have done a commendable job in recent years showing that outdated ideas of a microlith-assisted dispersal of Later Pleistocene humans from Africa is untenable because detailed comparisons of the African and Asian evidence show little technological resemblance (cf. Lewis, 2017. Early Microlithic Technologies and Behavioural Variability in Southern Africa and South Asia). Yet, in this manuscript, the authors use the presence of Levallois cores and flakes to link the Dhaba Locality assemblage to patterns seen

in the African MSA and thereby to dispersing *Homo sapiens*. The authors should discuss methods by which they could disprove the hypothesis that these technologies represent convergent evolution independent of an African source population. As currently framed, the argument cannot be falsified. In line 150 of the manuscript's supplementary material, the authors state, "A technological analysis of recently excavated artefacts from Patpara reveals the in situ emergence of Levallois reduction". Surely this statement runs directly in the face of the manuscript's central claim that Dhaba Locality's archaeology tracks "*Homo sapiens* as they dispersed eastward out of Africa."

As cited in the text, the authors have elsewhere shown that the core technologies of Middle Palaeolithic India, the Levant and Arabia (in the last two cases associated with modern human fossils), as well as Australia and SE Asia (Clarkson et al. 2012; Petraglia et al. 2007), as well as broader assemblage composition (Clarkson 2014), show much more in common with modern human assemblages from East African MSA than with Neanderthal assemblages from Europe and the Levant. The hypothesis is eminently testable since it would only require finding hominin skeletal remains in India of the same age as those at Dhaba to test the idea that modern humans or some other hominin/s were present at the time. Likewise, the archaeological comparisons are built on published quantitative data, and new data may change or alter those conclusions. Our sample does not comprise Denisovan assemblages for comparison since at this time it remains uncertain which assemblages from Denisova Cave or Tibet are in fact attributable to Denisovans, Neanderthals or modern humans. In any case, the lithic material from Denisova appears different from that of the sites located along the southern arc between Africa and Australia which all bare close similarity in quantitative comparisons.

The study cited in the supplementary information referring to in situ evolution of Levallois suggested that handaxes showed some similarities with Levallois cores, and Shipton et al. argued that handaxe technology could in theory evolve into Levallois given appropriate stimuli. However, there is no indication from the assemblage that handaxe technology actually evolved into Levallois at Patpara, nor is this meaning intended since the site is a single occupation layer. This line was removed from the SI.

In sum, the reviewer rightly noted a contradiction, and we have modified the SI for clarification.

169: It seems inappropriate to refer to the sequence as "spanning 55 thousand years" and not qualify the statement to the effect that the sequence is highly pulsed and punctuated.

We have modified this sentence to read:

"The stone artefact sequence at the three Dhaba excavations spans 55 thousand years, from 80 to ~25 ka with several pulses in artefact discard (Table S1)."

185: If micro is intended to mean small, can the authors explain why microblade cores > 4 cm are called micro (Supp Fig 6) while Levallois cores < 4 cm are not called micro-Levallois (Supp Fig 5)?

'Microlithic' is used in Supp Fig 6 to refer to an industry by that name arising around 45ka, not an individual stone artefact in this figure. Blade cores smaller than 4cm in length are referred to as microblades cores in the text. There is only one Levallois core smaller than 4cm. While it could be referred to as 'micro-Levallois' that has not been the convention in South Asian archaeology.

189: Can the authors clarify the meaning of proximal scar patterning?

Means simply that all scars originate from the proximal platform and run to the distal end. The caption to Figure 5 now includes these definitions:

“Proximal scar patterns originate from the proximal/platform end only. Strongly radial scar patterns originate in a centripetal pattern around the margins of the flake or core, while weakly radial scar patterns are those with scars that originate from several points around the flake or core circumference but are too few in number (<4) to determine a strong radial pattern.”

Table 1: Are these core types, flake types, both?

-What comprises the microlithic category in this table? Backed tools, bladelets, small flakes, small cores, bladelet cores? This seems an unnecessary catch-all category.

-Please explain how the percentages are calculated in this table (they don't add to 100 by stratum)?

-Why are backed artefact counts not presented here?

The composition of the table and the categories are now defined in Table caption as below. Backed artefact counts are not presented separately as there are too few to warrant a separate category.

Table 1 Composite stratigraphic sequence of counts and proportions of key types through time based on cores, flakes and retouched flakes. Microlithic pertains to counts and percentages of blades and microblades and cores as well as backed Microliths. Percentages are by stratum in each locality. Totals for Dhaba 3 are from squares 9-13.

Figure 4: The flake scar patterns (key for identifying Levallois technology) are terribly unclear on tools a, b, and c. The authors should include arrows indicating flake scar directionality on these figures. It is unclear how tool k is a point, it seems to be expanding in plan view, not converging, as one would expect for a pointed tool. The flake scar directionality is also not clear on this tool.

Figure 4 has been revised to include scar directions as well as impact points to indicate flake orientations so the points can be seen to contract rather than expand along their length.

Fig. 4 Key artefact types at Dhaba from 80 to 25 ka. **a-c** Levallois flakes, Dhaba 1 and 2. **d-e** Levallois blades, Dhaba 1. **f-g** ochre, Dhaba 1. **h-i** microblade cores, Dhaba 3. **j** notched scraper, Dhaba 1. **k-m** Levallois points, Dhaba 1 and 2. **n-o** agate and chert microblades, Dhaba 3. **p-s** recurrent Levallois cores, Dhaba 1-3. **t-u** backed microliths, Dhaba 3. White arrows indicate scar directions. Black arrows with circles indicate impact points.

Figure 5 a: Please define strongly and weakly radial so such classifications can be reproduced.

This is done in the caption to Figure 5 as above.

225: Can the authors provide some statistical measures for the “broad changes in raw material selection”. For example, is the trend through time significant?

Statistical comparisons are now provided for Fig 5 caption as per below for raw material and dorsal scar pattern changes:

Fig. 5 Key raw material and technological changes at Dhaba from 80 to 25 ka. **a** raw material changes as a proportion of all artefacts by stratum (Pearson Chi-Square = 2365; N = 90; p = <0.0005); **b** changes in proportions of scar patterning for all complete flakes by stratum (Pearson Chi-Square = 469; N = 342; p = <0.0005).

And for techno-typological changes, statistics were performed and included in the text, which now reads

“The broad changes in the proportions of key types through time shown in Table 1 are statistically significant (Pearson Chi-Square = 2109; N = 864; $p < 0.0005$).”

The statistical results for broad changes in technological characteristics are referenced in text to Clarkson et al. 2014 and 2018.

225: I did not see any data on changing retouch strategies, can the authors please highlight where this is?

This has been highlighted in the text. The sentence now reads:

“The technological changes in both the Middle Son and Jurreru River valleys appear to be stepwise and involve broad and statistically significant changes in raw material selection, changing retouch strategies (from scrapers and points to backed artefacts), systematic shifts in core reduction technology, and the introduction of new retouched artefact forms such as backed microliths as Levallois technology disappears^{31,32}.”

Reviewer 2

Overall, I support this paper. The geoarchaeology looks sound, and the team builds on previous work, and they are all highly experienced in this type of research, particularly in India. Those undertaking the OSL dating are in the top echelons of their field. In particular, the paper strengthens the case for lithic - and possibly population - continuity across the Toba ashfall of 74 ka. In the absence of skeletal evidence, it is the best that can be done at present.

I have a few additional comments:

page 2, line 3: "analysis of modern DNA indicates India was an important geographic stepping stone in the colonisation of Australasia by Homo sapiens". This needs a reference, e.g. Atkinson et al. 2008 Mol. Biol. Evol. 25(2):468–474.

p. 1 and 14, regarding Madjedbebe and Australian colonisation at 65 ka. The inclusion of this is controversial, and does not significantly strengthen their argument for a dispersal pre-80 ka. I can understand why Clarkson wishes to include it as he was the lead author on the Nature paper. Nevertheless, it has been criticised, especially by O'Connell et al. PNAS 2018, and as far as I know, Clarkson et al. have not responded to the suggestion of extensive bioturbation. Personally, I think it should be omitted from the paper.

p. 15: "Modern human dispersal out of Africa must, therefore, have taken place before 65 ka". Two points: first, they should include the recent skeletal evidence from Misliya, Israel, dated at 177-194 ka. Second, the issue here is not "human dispersal out of Africa" because we have the skeletal evidence from Misliya, Skuhl, Qafzeh and Wusta that all show that sapiens was out of Africa before 65 ka. The issue is not dispersal out of Africa, but east of Arabia - it is that part of the Out of Africa debate that is contentious. I suggest rephrase here as suggested.

In the SI, I support their argument that there is considerable spatial variability along rivers such as the Son. Although I have not seen the Son, I have observed similar variability along South Asian rivers and palaeo-rivers, e.g. the Narmada.

Please find our responses below

page 2, line 3: "analysis of modern DNA indicates India was an important geographic stepping stone in the colonisation of Australasia by *Homo sapiens*". This needs a reference, e.g. Atkinson et al. 2008 *Mol. Biol. Evol.* 25(2):468–474.

This reference is now added

p. 1 and 14, regarding Madjedbebe and Australian colonisation at 65 ka. The inclusion of this is controversial, and does not significantly strengthen their argument for a dispersal pre-80 ka. I can understand why Clarkson wishes to include it as he was the lead author on the *Nature* paper. Nevertheless, it has been criticised, especially by O'Connell et al. *PNAS* 2018, and as far as I know, Clarkson et al. have not responded to the suggestion of extensive bioturbation. Personally, I think it should be omitted from the paper.

These are published ages for a site that many do accept. We are simply citing the published literature and outlining how our results fit with current patterns worldwide. Therefore we see no reason not to include reference to this site and to Clarkson et al.'s *Nature* paper. O'Connell et al.'s paper in *PNAS* is an opinion piece, and it includes no new data, and merely conjectures as to site formation processes. In any case, their supposition is addressed and refuted by Smith et al. in the journal *Geoarchaeology* in 2019 (see Smith, M.A., Ward, I. and Moffat, I., How do we distinguish termite stone lines from artefact horizons? A challenge for geoarchaeology in tropical Australia).

p. 15: "Modern human dispersal out of Africa must, therefore, have taken place before 65 ka". Two points: first, they should include the recent skeletal evidence from Misliya, Israel, dated at 177-194 ka. Second, the issue here is not "human dispersal out of Africa" because we have the skeletal evidence from Misliya, Skuhl, Qafzeh and Wusta that all show that sapiens was out of Africa before 65 ka. The issue is not dispersal out of Africa, but east of Arabia - it is that part of the Out of Africa debate that is contentious. I suggest rephrase here as suggested.

This is a good point and we have included it. This final two sentences now read:

"Modern human dispersal out of Africa, and more importantly east of Arabia, must therefore have taken place before ~65 ka, so cultural and fossil evidence from sites dating to this period will be important for future tests of this hypothesis, notwithstanding the fact that population contractions and turnovers may have also occurred. The Dhaba locality serves as an important bridge linking regions with similar archaeology to the east and west."

Reviewer 3

I was asked to review this paper "*specifically looking for advice on the tephrochronology aspect of this study*" which I have considered. To set this paper in context, the luminescence dating places a series of fluvial deposits from the Son Valley in the time frame of the YTT eruption, and this time is important in hominid migration etc. Many studies have located and Identified YTT in India – the

authors cite a limited number of these studies, missing papers in 2013, 2014, and 2017 by Westgate and colleagues which provide analytical data for YTT in India, and mechanisms for recognising YTT.

Terminology: the authors refer to “Toba ash”, “Toba super-eruption” etc. in many places. The Toba Caldera Complex in Sumatra underwent several super-eruptions (OTT, MTT and YTT) thus “Toba ash” is equivocal, and when these terms are used, they should refer to the specific eruption being discussed. There have been suggestion that OTT occurs in India, which have been discounted, but clarity is required when referring to a particular Toba eruption.

Review: Reading the title, which includes “Toba ash”, I was expecting to see samples from a site with a visible tephra layer (which I presumed would be YTT – see comment above) and analyses to confirm the identification of the layer. As I read the paper, the term cryptotephra was used (Line 38, which clearly indicates “*above and below cryptotephra derived from the Toba volcanic super-eruption (Youngest Toba Tuff or YTT) at ~74 ka.*” (again, this is not “**the** Toba super-eruption”, as there were several, but “the YTT super-eruption”). So, not a visible layer, but a cryptotephra layer, but this should provide an isochron, and surely there will be some data or plots of chemistry later in the paper to confirm that headline identification as YTT.

Line 78 finally tells us for the first time about the tephra “layer” describing “*the presence of five tephra glass shards, attributed to the Toba ash*” and again five shards get mentioned on Line 155, but when you look at Figure 3, you count six shards (2+2+1+1) from 4 samples. Is it five or is it six shards? Whatever the number, it is an alarmingly sparse tephra deposit.

Where then is analytical data to confirm the attribution of these few shards to YTT? Are there major element analyses by EPMA which may tie the material to Toba, and perhaps also trace element analyses, the only robust way to identify it as YTT, and distinguish it from OTT (not yet recorded in India, but many argue the possibility that it could be present)? There is no mention of any analyses in the section describing “Cryptotephra shards” (lines 154-166), but mention here of many sites in the vicinity where YTT occurs. The paper continues.....“*The stratigraphic position of the Dhaba 1 shards is consistent with the IRSL ages of 78–80 ka for the lower unit and 65–70 ka for the upper unit, which suggests that the shards are in primary context or were reworked into the overlying deposits*” This sentence implies the shards have been identified and are now in the correct stratigraphic position (based on the OSL ages), but where is the data to show these are YTT?

There is a hint of circularity to this – these shards, which the authors have already said are YTT, are in the right place, because the ages tell us they are Toba? As yet, no analytical data has been presented to suggest these are even rhyolite, let alone material derived from Toba, or even specifically YTT.

In **Methods**, for Cryptotephra (Lines 263 – 271), again no data are represented to justify the assignment to YTT, but the consistency of the ages of the sediments with the eruption age of YTT is used to justify the stratigraphic context, and assignment of a YTT origin. Again, no evidence is presented to establish the identity of the glass by methods which are extensively used in tephra and cryptotephra studies.

Eventually, residing in the **Supplementary information**, you find some more detail of the “*Cryptotephra investigation*”. Standard methods were used to prepare and separate samples hopefully containing glass, and mounted to identify shards microscopically. A few glass shards were found in total, but crucially the glass shards were not analysed “*because they were too low in abundance*”. In the S.I. the paper states “*our correlation to the YTT is based upon the demonstrated presence of YTT within the Son river valley*” – so not an assignment based on compositional data or

any other measurable parameter (perhaps they could try RI data which would at least tell you they are/are not rhyolitic), but assumed to be YTT simply because there is some near-by. The OSL ages and the “visual appearance” of the shards are used back this up. The authors also cite the fact that no other tephra apart from YTT has been identified in the area, as further justification (but miss 3 or 4 papers which are post-2011 relating to YTT in India).

The “recognition” of YTT in these samples is thus based on an assumption, and is not identification, and should not be used. Further material should have been prepared so enough glass could be obtained for analysis (this is presumably still possible?), and then (and only then) can a reliable identification be attempted. The headline presence of “Toba tephra” in the title is misleading – and whilst it I would not be at all surprised if this does turn out to be YTT, at present there is no evidence whatsoever to support this.

Summary: Any mention of a Toba/YTT assignment to the cryptotephra should be removed from the paper entirely; it is misleading and not proven. In the context of other sites in the area where there are clear layers of tephra the title leads you to imagine something that is not there, and these misconceptions stick – in 2 years’ time I can imagine a paper appearing which quotes “Clarkson et al. (2019) identified YTT in sediments from Dhaba, India”and there it is, forever entrenched in the literature, when there has actually been no robust identification of the glass.

A more open approach should be adopted in the paper, along the lines of “Five [*or is it six?*] uncharacterised, colourless [*I’m not sure if they state the colour*], platy volcanic glass shards were recovered from four samples taken for luminescence dating. These were at very low concentrations (<1 shard/cm³), and it was not possible to analyse these. As yet, their source volcano cannot be confirmed”, and no further speculation about what they are should be made. It is appropriate to refer to YTT in many other sites across India, but the reader can be left to speculate on whether these are the same – it should not be fed to them without proper evidence to support it. Furthermore, “and Toba ash” should not appear in the title, nor Toba/YTT etc in the abstract or text, figures etc., instead “unidentified glass shards” should be used until it can be proved using standard analytical approaches that these are indeed from the YTT eruption, or not.

Recommendation: The paper needs modification to remove any mention of these shards providing a justification for the stratigraphic context of the artefacts, and needs to be rephrased throughout so as not to indicate their provenance, until this is confirmed by appropriate analytical means.

Reviewer 3 questions whether the **six** glass shards found at the site were sourced from the Youngest Toba Tuff (YTT) volcanic eruption ~74,000 years ago, given that we have been unable to chemically characterise the shards. In the revised manuscript, we have adopted a conservative approach to our assessment of the source of the glass shards as recommended by this reviewer, explicitly declaring that we cannot identify the source chemically, while also noting the following:

- (1) all known occurrences of Toba ash in India have been chemically correlated with the YTT event, which is known to have deposited large quantities of ash across India;
- (2) large quantities of chemically fingerprinted and directly dated YTT deposits have been thoroughly documented at the site of Ghogara, which is located only 700 m from Dhaba (we have now added an image of the Ghogara section to show the extensive ash deposits found *in situ* there); and
- (3) the glass shards at the study site were found in deposits dated to a time period consistent with that of the YTT eruption.

We have now removed all mention that the six YTT shards at Dhaba are definitively associated with the YTT event, restricting ourselves to saying that it is the most likely source, given these three supporting lines of evidence.

Crucially, the chemical identification of the few glass shards found at Dhaba is not required to uphold our key finding that human occupation at the site spans the 74,000 year-old Toba eruption. The luminescence ages of between 78,000 and 65,000 years for the artefact-bearing deposits at Dhaba provide independent chronological support for human occupation of this site before and after the YTT event. Hence, the presence of the YTT shards at Dhaba is merely “icing on the cake”, as it were – we would have drawn the exact same conclusion, that people occupied the site before and after the YTT event, even in the absence of glass shards in the Dhaba deposits.

Reviewers' Comments:

Reviewer #1:

Remarks to the Author:

I am satisfied with the authors response to my comments and I would recommend that the paper be published without further revisions.

Reviewer #3:

Remarks to the Author:

I have reviewed the changes made in the paper by Clarkson et al and am pleased to see the title no longer includes reference to Toba ash. Unfortunately, no further samples have been prepared, so the data upon which their tephra identification is based remains unchanged. Thus, the author continue with the supposition that these glass shards must be YTT, based on the occurrence of YTT nearby at other sites in the Son Valley and their dating, but not on any analytical or other evidence to support the claim that YTT glass is present at the Dhaba sites. The supplementary information on tephra processing etc is unchanged.

The modifications the authors have made to the paper have however brought into focus another question related to the occurrence and method of collection of these glass shards.

Whist the inconsistencies as to the number of shards discovered (6 in total) have been cleared up, this still remains a woefully small amount of material upon which to claim the presence of an isochron – these 6 shards were recovered from 4 out of 32 2-3 g samples which were processed for cryptotephra analysis. Importantly, no detail is given of how these samples were collected. Why were they were so small – they cannot have been much more than 1 cubic cm in volume? I wonder if these were surface samples taken for palaeomag work in small plastic boxes, which would be about this size, or were they surface scrapes from the trench (see Haslam et al 2012 Quat Int, Fig 4c) or sub-samples from something bigger? This needs to be clear as it has significant implications (see below). Whatever, I cannot understand why the authors did not prepare more material for tephra analysis, for example, from the luminescence samples, where they will have obtained a date and any tephra would go hand-in-hand. I appreciate the difficulty in the cryptotephra sample preparation, but the lengths the authors go to attempting to convince the reader the glass is YTT (in the absence of any chemical evidence), surely an approach to obtain more material would remove any doubt surrounding this assignment. My initial review centred largely on this lack of proof (analyses) that this material was YTT. This concern still stands.

A further problem only became apparent from the revisions made to the m/s in an attempt to justify the presence of near-by YTT, and is really a blindingly obvious problem. In revising the m/s the authors provided more detail on other occurrences of YTT in India (lines 175-185) to support their assumption that the glass shards found at Dhaba are YTT. In doing this, they now include photos (Fig 3) from Emma Gatti, who worked a section at Goghara (700 metres from Dhaba) which contains a thick YTT layer. This is interesting because

- the acknowledgements in Gattis's PhD thesis (2012, Cambridge Univ.) indicate two of the present authors (Harris and Neudorf) helped in her fieldwork
- Haslam et al (2012, Quat Int) detail preliminary filed work at Dhaba in a paper including five of the authors on the current submission, and thank "Christina Neudorf, Emma Gatti, Adam Durant and the villagers of Dhaba for their contributions in the field", with Fig 4d showing a field party of about a

dozen

- Neudorf et al (2014, Palaeo³) published a paper on OSL dating at the Goghara site (the same site worked by Gatti and shown in Figure 3) and Neudorf thank “the local villagers of the MSV for help with field excavations and sampling” in their acknowledgements.

It appears from these works that a large field party (including PhD students, locals, academics) undertook the excavations in the Son Valley, with at least some working both the Ghogara and Dhaba sites, and when only 700 m separates the sites, it is hard to imagine that most people did not work/visit both sites. However, any individual who worked at Ghogara will almost literally have been bathed in YTT glass from the “thick” deposit exposed there, and if anyone moved to the nearby Dhaba site (where there is no exposed tephra) during the same field season, the risk of contamination between sites (from people, tools, clothing, field gear etc etc) is enormous, if not essentially certain. Cryptotephra studies from peat/lake/ice cores etc, or sequences where there is no visible tephra exposed (essentially “clean” sequences) are at very low/no risk from this type of contamination, but where there is such a contrast in the abundance of glass in close proximity in surface sites, the risk is significant. When the claimed tephra “layer” at Dhaba comprises only 6 shards recovered from 4/32 small samples, I am extremely surprised that none of the authors on the current submission have considered and addressed the possibility that these shards are the result of accidental contamination of the Dhaba site by people working there and at Ghogara.

The authors must categorically address this risk of contamination, and provide more detail of exactly what was sampled for the tephra analysis. They must be open about the “traffic” between the Dhaba and Ghogara sites (numbers of people, timing of sampling etc) and assess (honestly) the risk of their glass shards being contaminants. I would not doubt that glassy material has been moved between sites where you have such a contrast in the amounts of tephra (a 1 metre thick layer and none), and for this reason I would not include any mention of the glass, because there cannot be certainty it is in situ. A precautionary approach would surely accept the possibility that these shards are simply inter-site contamination, until more evidence (i.e. more shards from more secure samples) can be presented.

Of course, as described for other sites, the presence of YTT alongside archaeological finds can make the dating more secure, and adds significant weight to the archaeological argument. With the present authors well versed in archaeo-tephra studies in India, I am left to wonder whether the possible implications of sample contamination have been overlooked in favour of the security that even the most tenuous claim for the presence of YTT tephra layer provides the archaeology.

My suggestion would be to ignore the presence of the glass at Dhaba completely, and work the paper up on the archaeology/dating alone which is secure and provides dates which bracket the YTT eruption – I fear the authors must admit to the possibility that those 6 shards are contaminant material moved between sites during such a big field sampling campaign: this cannot be excluded.

Response to reviewers

Reviewer 3 (and Reviewer 1 is in agreement) has requested further tephra studies or that that any mention of the cryptotephra is further caveated or even removed. Since we are unable to perform further geochemical studies on the tephra, we have revised the main text significantly to remove any assertions that the cryptotephra is YTT, or that it forms an isochron through the Dhaba 1 stratigraphy.

We have removed the following sentence from the introduction:

Given the occurrence of thick deposits of volcanic ash ~700 m from the site and chemically identified as derived from the eruption of the Toba volcano in northern Sumatra ~74 ka (i.e., the Youngest Toba Tuff, YTT), the source of all known Late Pleistocene tephra deposits in India²⁻¹⁰, we consider the Dhaba shards are most likely also related to the YTT event.

And replaced this with:

An unchanging stone tool industry is found at Dhaba spanning the Toba eruption of ~74 ka (i.e., the Youngest Toba Tuff, YTT)²⁻¹⁰ bracketed between ages of 79.6 ± 3.2 and 65.2 ± 3.1 ka, with the introduction of microlithic technology ~48 ka.

We have now added a short paragraph at the end of the IRSL dating section discussing the presence of the six shards but stating that its origin is uncertain:

Interestingly, six glass shards were found at Dhaba 1 in deposits dated to between 79.6 ± 3.2 and 65.2 ± 3.1 ka (Fig. 3a, see Supplementary Information and Supplementary Table S4), which is consistent with the known date of the YTT event and the widespread presence of YTT in India and the Middle Son Valley^{2-10,30}. However, we cannot rule out contamination by human agency as a possible source of these few shards at Dhaba 1, given the presence of thick YTT deposits at nearby sites that were visited by some of the same researchers.

We have modified the final paragraph of the supplementary information to the same effect.

While these shards are likely YTT, based upon the demonstrated presence of YTT within the Son River valley (Westgate et al., 1998; Jones 2007; Jones and Pal, 2009; Gatti et al., 2011; Smith et al., 2011), chronological fit with the OSL ages of the sediment (this study) and correspondence of the visual appearance of the glass shards, we cannot rule out the possibility that these few shards are the result of human contamination from researchers working at both Dhaba 1 and the nearby site of Ghogara, where YTT deposits are abundant (Gatti et al. 2011).